# Coupling threshold theory and satellite image derived channel width to estimate the formative discharge of Himalayan Foreland rivers.

Kumar Gaurav[1], François Métivier[2], AV Sreejith[3], Rajiv Sinha[4], Amit Kumar[1], and Sampat Kumar Tandon[1]

[1]Indian Institute of Science Education and Research, Bhopal,462066, M.P, India
[2]Institute de Physique du Globe de Paris, 1 Rue Jussieu, 75005 Paris cedex 05, France
[3]School of Mathematics and Computer Science, Indian Institute of Technology, Goa, 403401, Goa, India
[4]Department of Earth Sciences, Indian Institute of Technology, Kanpur, 208016 UP, India

**Correspondence:** K.Gaurav (kgaurav@iiserb.ac.in)

**Abstract.** We propose an innovative methodology to estimate the formative discharge of alluvial rivers from remote sensing images. This procedure involves automatic extraction of the width of a channel from Landsat Thematic Mapper, Landsat 8, and Sentinel-1 satellite images. We translate the channel width extracted from satellite images to discharge by using a width-discharge regime curve established previously by us for the Himalayan Rivers. This regime curve is based on the threshold theory, a simple physical force balance that explains the first-order geometry of alluvial channels. Using this procedure, we estimate the formative discharge of six major rivers of the Himalayan Foreland: the Brahmaputra, Chenab, Ganga, Indus, Kosi, and Teesta rivers. Except highly regulated rivers (Indus and Chenab), our estimates of the discharge from satellite images can be compared with the mean annual discharge obtained from historical records of gauging stations. We have shown that this procedure applies both to braided and single-thread rivers over a large territory. Further our methodology to estimate discharge from remote sensing images does not rely on continuous ground calibration.

**Keywords:** Himalayan Foreland; regime curve; threshold theory; formative discharge

## 1 Introduction

The measurement of river discharge is necessary to investigate channel morphology, sediment transport, flood risks, and to assess water resources. Despite this, the discharge of many rivers remains unknown, especially those located in sparsely populated regions, at high latitudes, or in developing countries. Even now, the discharge is measured at sparsely located stations along a river's course (Smith and Pavelsky, 2008; Andreadis et al., 2007). Between measurement stations, the discharge is interpolated using routine techniques (Smith and Pavelsky, 2008). Further, these local measurement stations are installed where the river flows as a single-thread channel and has a stable boundary. This is often not the case for braided rivers, where the flow

is distributed through multiple and mobile threads (Smith et al., 1996; Ashmore and Sauks, 2006). Braided rivers are therefore often not gauged; and where these exist, the gauging stations are located at places like dams with artificially regulated flow. This hinders our ability to assess discharge in the individual threads of a braided river.

To overcome this problem, and to minimise the costs related to discharge measurement, methodologies have been developed to use remote sensed images to estimate the instantaneous discharge of rivers (Smith et al., 1996; Smith, 1997; Alsdorf et al., 2000; Ashmore and Sauks, 2006; Alsdorf et al., 2007; Marcus and Fonstad, 2008; Papa et al., 2010, 2012; Gleason and Smith, 2014; Durand et al., 2016; Gleason et al., 2018; Allen and Pavelsky, 2018; Moramarco et al., 2019; Kebede et al., 2020). These studies establish rating relationships between some image-derived parameters (width, water level or stage, slope), to the instantaneous discharge measured in the field (Leopold and Maddock, 1953). Equations that define the hydraulic geometry of a channel relate width (W), average depth (H), and slope (S) of a channel to the bankfull discharge (Q) according to:

$$W = aQ^e, \tag{1}$$
$$H = bQ^f, \tag{2}$$
$$S = cQ^{-g}, \tag{3}$$

where $a, b, c, e, f, g$ are site specific constants and exponents. The available methods, based on remote sensing data, to estimate the discharge of a river therefore cannot be extrapolated to other rivers, or even to other locations on the same river. Moreover, as these rating curves vary significantly between locations, they must be established for each location independently. For example, Smith et al. (1995); Smith (1997); Smith and Pavelsky (2008) and Ashmore and Sauks (2006), used synthetic aperture radar and ortho-rectified aerial images to estimate discharge in braided rivers. They related the image derived effective width of a braided river to the discharge at a nearby gauge station to establish a relationship of the form of equations 1, 2, & 3. Their approach provides an estimate of the total discharge in a braided river, at a given section. However, this technique is site specific and assumes that the river bed does not change over time.

Few attempts have been made to overcome these limitations; for example Bjerklie et al. (2005) used aerial orthophotographs and SAR images to estimate discharge in various single-thread and braided rivers. To estimate the discharge they extracted the maximum water width at a given river reach. They then combined the image-derived channel widths with channel slopes obtained from topographic maps, and a statistical hydrologic model. They reported standard errors of $50 - 100\%$. However, after using a calibration function based on field observation, the error reduced to values as low as $10\%$. Later, Sun et al. (2010) used Japan Earth Resource Satellite-1 (JERS-1) SAR images to measure the effective width of the Mekong River at the Pakse gauging station in Laos. They used rainfall-runoff model to estimate the discharge from the image-derived width and suggested that using this procedure, the discharge could be estimated in any ungauged river basin within an acceptable level of accuracy. They established a close agreement between the measured discharge of the Mekong River at Paske station and the model estimate to the $90\%$ uncertainty level. As discussed earlier by Bjerklie et al. (2005), later Sun et al. (2010) indicated that the precision can be improved by calibrating the rainfall-runoff model with a hydraulic geometry relation, and that a calibrated rainfall-runoff model can be used to estimate the discharge in any ungauged river using the measured width only. Gleason and

Smith (2014) have suggested that the discharge of a single-thread river can be estimated from satellite images only, without any ground measurement. They plotted the exponents and coefficients of hydraulic regime equations established at 88 different gauging stations along six rivers in the United States, and found that the exponents and coefficients are correlated. Recently Kebede et al. (2020) have used Landsat images to estimate daily discharge of the Lhasa River in the Tibetan Plateau. They have used image derived hydraulic variables to compute the discharge by using modified Manning equation and rating curves established from the in-situ measurement of width and discharge.

The studies discussed above attempt to address the issue of site-specificity, and propose methods to estimate discharge without empirical calibration. However Bjerklie et al. (2005), and Sun et al. (2010) also show that a better accuracy in discharge prediction can only be achieved with some calibration to ground measurements. Therefore, a physically robust method to resolve the site-specificity of rating curves remains to be described.

To address this issue of site-specificity, we have developed a semi-empirical width-discharge regime relation based on the threshold theory and field measurement of various braided and meandering rivers on the Ganga and Brahmaputra plain (Seizilles et al., 2013; Métivier et al., 2016; Gaurav et al., 2017). According to this relation, threads of braided and meandering rivers share a common width-discharge regime relationship. We therefore hypothesise that, this regime equation can be used to estimate the first order discharge of any river (braided or meandering) flowing on the Ganga and Brahmaputra plains, and perhaps on the entire Himalayan Foreland, if wetted width of the river channels is known. This study can also be used for various applications such as: (i) to monitor the downstream evolution of discharge, (ii) to fill the data gap in between the gauge stations separated over a long distance, (iii) to construct the time-series and trend analysis of discharge variation, and (iv) to identify the critical reaches in rivers that are under stress due to excessive extraction of water for agriculture, industrial or domestic supply.

## 2   Hydrology of the Himalayan Rivers

Many rivers flowing on the Indus-Ganga-Brahmaputra alluvial plains are perennial and have their source in the Himalaya and Tibetan Plateau. Flow of these rivers is primarily determined by snowmelt and rainfall during the Indian summer monsoon (Singh and Jain, 2002; Thayyen and Gergan, 2010; Bookhagen and Burbank, 2010; Andermann et al., 2012; Khan et al., 2017). However, the contribution of rainfall and snowmelt in the discharge of the Himalayan rivers vary significantly along the orogenic strike. For example, on an annual timescale, snowmelt contributes about 15-60% of discharge in the western Himalayan rivers, whereas it is less than 20% in the eastern Himalayan rivers (Bookhagen and Burbank, 2010). These rivers experience a strong seasonal variability in their discharge, for instance rainfall during the Indian summer monsoon (June-September) constitutes about 60-85% of the eastern and about 50% of the annual discharge of the western Himalayan catchments.

A closer look into the hydrographs of the Himalayan rivers reveals two distinct flow regimes (Fig. 1). A clear separation of discharge during the summer monsoon and rest of the period can be observed. From May to October, most of the Himalayan rivers flow at their peak discharge due to intense and prolonged rainfall and glacier melting in the catchment; whereas, in lean period (November-April), they carry relatively less discharge.

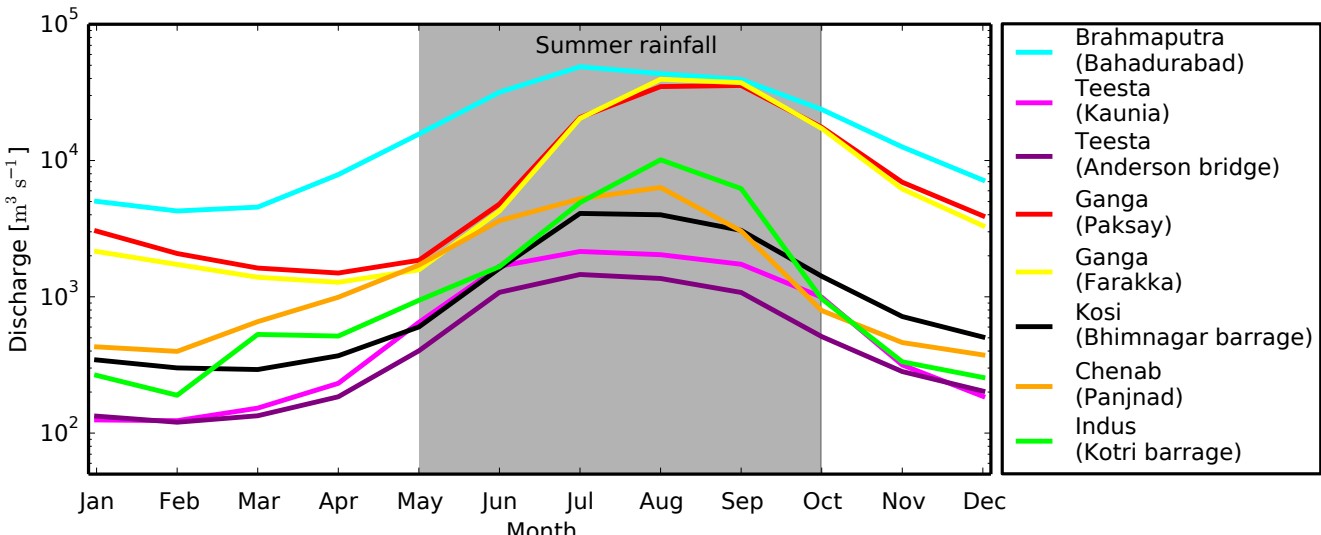

**Figure 1.** Hydrograph of the Himalayan Rivers

## 3 Morphology of alluvial river

Lacey (1930) was the first to observe a dependency of width of an alluvial river on its discharge. Based on measurements in various single-thread alluvial rivers and canals in India and Egypt, he found that the width of a regime channel scales as the square root to the discharge ($e \sim 0.5$ in Eq. 1).

To explore the physical basis of Lacey's observation, Glover and Florey (1951) and Henderson (1963) developed a theory based on the concept of threshold channel. According to this theory, with a constant water discharge, the balance between gravity and fluid friction maintains the sediment at threshold of motion, everywhere on the bed surface. This mechanism sets the cross-section shape and size of a channel. The resulting width ($W$) - discharge ($Q_w$) relationship in dimensionless form reads (Seizilles, 2013; Gaurav et al., 2014; Métivier et al., 2016, 2017; Gaurav et al., 2017):

$$\frac{W}{d_s} = \left[ \frac{\pi}{\mu} \left( \frac{\theta_t (\rho_s - \rho_f)}{\rho_f} \right)^{0.25} \sqrt{\frac{3C_f}{2^{3/2} \mathcal{K}[1/2]}} \right] Q_*^{0.5} \tag{4}$$

where $Q_* = Q_w / (d_s^2 \sqrt{g d_s})$ is the dimensionless water discharge, $d_s$ is the grain size, $\rho_f \approx 1000 \, \mathrm{kg \, m^{-3}}$ is the density of water, $\rho_s \approx 2650 \, \mathrm{kg \, m^{-3}}$ is the density of quartz, $g \approx 9.81 \, \mathrm{m \, s^{-2}}$ is the acceleration of gravity, $C_f \approx 0.1$ is the Chézy friction factor, $\mu \approx 0.7$ is the Coulomb's coefficient of friction, $\mathcal{K}(1/2) \approx 1.85$ is the elliptic integral of the first kind, and $\theta_t$ is the threshold Shield's parameter that depends on the sediments grain size. The typical grain size of the sediments of the Himalayan Foreland rivers is order of $d_s = 100 - 300 \, \mu m$. Thus the dimensionless grain size $D^* = (d_s^3 g \rho_s^2 / \eta^2)^{1/3} \simeq 1 - 6$, where $\eta \approx 10^{-3} \, \mathrm{Pa.s}$ is the dynamic viscosity of water. In this range of values the threshold Shield number is on order of $\theta_t \sim 0.1$ with a maximum around 0.3 (Julien, 1995; Selim Yalin, 1992). Recently Delorme et al. (2017), obtained an experimental value of $\theta_t \sim 0.25$ for silica sands of size $150 \, \mu m$. Here we have taken the upper value of $\theta_t = 0.3$ as a conservative estimate.

Taking lower values of threshold Shield parameter, such as the classical 0.1 would lead to a slightly better match between the theoretical prediction and the data but it does not lead to a significant change in our conclusions.

  Eq. 4 is the theoretical equivalent to the Lacey's law. This theory explains the mechanism how a single-thread alluvial river, at threshold of sediment transport, adjust their geometry in response to the imposed water discharge. Strictly speaking, mean equilibrium geometry of a natural alluvial channel is not set by a single discharge, rather a range of discharges is responsible
for determining the channel form (Leopold and Maddock, 1953; Wolman and Miller, 1960; Blom et al., 2017; Dunne and Jerolmack, 2020). However, what value corresponds to the channel forming discharge of an alluvial river remains a matter of debate. Wolman and Miller (1960); Wolman and Leopold (1957); Phillips and Jerolmack (2016) proposed that the bankfull discharge and discharge associated with a certain frequency distribution can be used to define the channel forming discharge.

  Since threshold theory predicts the scaling relationship of a single-thread channel, one may consider applying it to assess
the discharge that relates to present day geometry of natural alluvial channels. To test this, we use the regime curve that we established from threshold theory and measurement of hydraulic geometry of various sandy alluvial rivers in the Himalayan Foreland (Gaurav et al., 2014, 2017). In field campaigns during 2012, 2013, 2014, and 2018 we measured the geometry (width, depth, velocity and median grain size) of individual threads of braided and meandering river spanning over the Ganga and Brahmaputra plains. To measure the channel geometry we have used Acoustic Doppler Current Profiler (ADCP) on an
inflatable motor boat. Close to the location of our ADCP-measured transects, we collected the sediment sample from the channel. We sieved the sediment sample in the laboratory to calculate the median grain size ($d_{50}$). A detailed descriptions of the measurement can be found in our previous publications (Gaurav et al., 2014, 2017).

  Figure 2 suggests that the individual thread of the Himalayan Foreland rivers share a common width-discharge regime relation, and to the first order their morphology can be explained by threshold theory. The theoretical exponent accords with
125 the empirical exponent of the width-discharge curve. However, the threads are wider than predicted by a factor of about 2 (Fig. 2). We now adjust the prefactor predicted from threshold theory to our data while keeping the theoretical exponent to establish a generalised semi-empirical "width-discharge" regime relationship for the Himalayan Foreland rivers (Fig.2). We then use this curve to estimate the discharge of various rivers of the Himalayan Foreland by measuring their width from satellite images.

## 4 Material and method

### 4.1 Dataset

To measure the width of a river channel, we use images acquired from Landsat Thematic Mapper (TM), Landsat 8 and Sentinel 1A satellites (Appendix A1). All images of the Landsat and Sentinel satellite missions are freely available and they can be downloaded from the US Geological Survey (https://earthexplorer.usgs.gov) and Alaska Satellite Facility (https://www.asf.alaska.edu/sentinel) websites. We have downloaded all available cloud-free Landsat satellite images, at the locations that were
135 near the in-situ measurement stations for which discharge data was available with us (Fig. 3). Only a few cloud free Landsat images are available for the period of June to September. This is mainly because of the strong monsoon that causes intense rainfall and dense cloud covers. To overcome seasonal effect and fill the data gap during the monsoon period, we use Sentinel

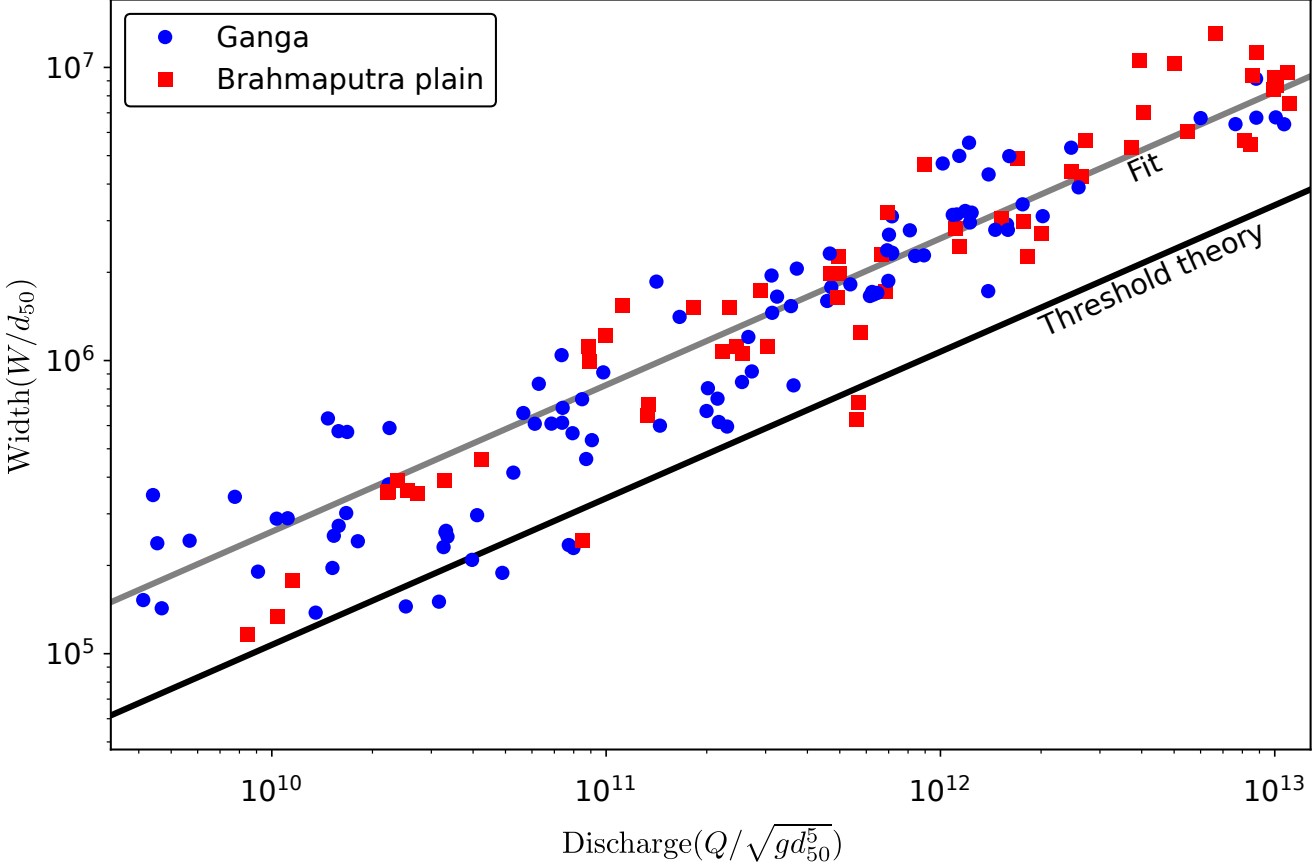

**Figure 2.** Dimensionless width of the individual threads of the Himalayan Foreland rivers as a function of dimensionless water discharge (after: Gaurav et al. (2017)). These data (width, discharge and grain size) were acquired during the different field campaign in years 2012, 2013, 2014 and 2018. The measurement was performed during the period when the Himalayan river usually flow at their formative discharge. The solid line (dark) is the prediction from threshold theory and the solid line (light) is obtained by fitting the prefactor of the threshold relation (Eq.4) to the data while keeping the theoretical exponent.

1A product. Sentinel-1 satellite mission is equipped with Advanced Synthetic Aperture Radar (ASAR) sensor that operates in C-band (5.4 GHz) of microwave frequency (Schlaffer et al., 2015; Martinis et al., 2018). Advanced Synthetic Aperture Radar system can operate both day and night and has the capability to penetrate clouds and heavy rainfall. This special characteristic of SAR sensors enables uninterrupted imaging of the Earth's surface during the bad weather conditions as well.

In-situ measurements of average monthly discharge for some time intervals of varying length between 1949-1975 are available for the Brahmaputra, Teesta, Ganga, Chenab, and Indus rivers of the Himalayan Foreland. They can be freely downloaded from (http://www.rivdis.sr.unh.edu/maps). We could obtain discharge data for the period 1996 - 2005, for the Ganga River at Paksay station and the Brahmaputra River at Bahadurabad station from Bangladesh. Similarly, the Ganga River discharge from

1978 - 2007, measured at the Farakka station in India was obtained from the Central Water Commission, Ministry of Water resources, New Delhi. We also obtained discharge data for the Kosi River for the period 2002 - 2014, from the investigation and research division, Kosi project, Birpur and from our own field measurements (Appendix A2). We obtained the median grain size of the bed sediments of the Kosi, Teesta and Ganga rivers from our own measurements in the field, whereas for the Chenab, Indus and Brahmaputra rivers from the published literature (Goswami, 1985; Dade and Friend, 1998; Gaurav et al., 2017; Khan et al., 2019). The median grain size ($d_{50}$) of our rivers vary in a narrow range between $250 - 115\,\mu m$.

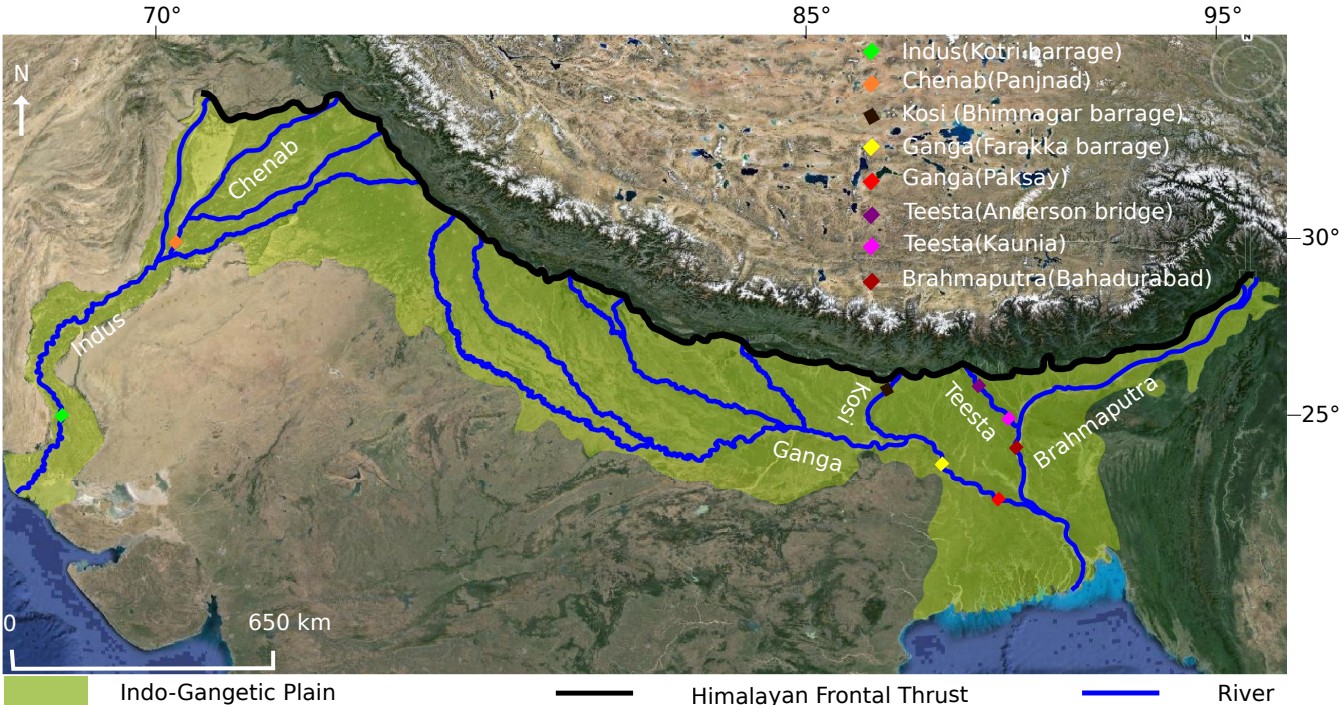

**Figure 3.** Location of the gauge stations of various rivers on the Indus, Ganga and Brahmaputra basins for which discharge data is available (source: https://www.google.com/earth ©Google Earth).

## 4.2 Width extraction

Our main objective is to extract the width of individual river channels from satellite images. We have developed an automated program in python 3.7 that takes a gray scale image as an input to classify the image pixels into binary water and non-water classes. The pixels classified as water are the foreground object and will be used to define river channels. Dry pixels serve as a background object. To extract the river channels, we use the infra-red bands of Landsat-TM and Landsat-8 images. In Landsat-TM, the infra-red ($0.76 - 0.90\,\mu m$) wavelength corresponds to band 4 whereas, in Landsat-8 image, it corresponds to band 5 ($0.85 - 0.88\,\mu m$). The spatial resolution of infra-red band for both Landsat- (TM & 8) missions is 30 m (pixel size: $30 \times 30$ m). Theoretically, since water absorbs most of the infra-red radiations it appears dark, with an associated brightness

value close to 0. This typical characteristic of the infra-red signal allows a clear distinction between the water covered dry areas on the satellite images (Frazier et al., 2000). However, in the case of a river, the pixel intensity varies widely because of heterogeneous reflectance of river water, due to the presence of sediment and organic particles (Nykanen et al., 1998). Because the image intensity is not exactly 0 or 1, we introduce a threshold intensity to classify the pixels. Based on this criteria, we convert the gray scale image $f(x, y)$ into a binary image $g(x, y)$, which distinguished between the water-covered and dry areas.

This approach takes an object-background image and selects a threshold value that segments image pixels into either object (1) or background (0) (Ridler and Calvard, 1978; Sezgin et al., 2004).

$$g(x,y) = \begin{cases} 0, & \text{if } f(x,y) < T \\ 1, & \text{if } f(x,y) >= T \end{cases} \tag{5}$$

We apply the algorithm proposed by Yanni and Horne (1994) to obtain the threshold value iteratively. Once this optimal value is obtained, we apply it to classify our pixels into water and dry classes (Fig. 4). The binary classification of satellite images

into water and dry pixels can produce spurious features as well (Fig. 4). These consist of wet pixels that get classified as dry or of isolated water pixels that appear randomly in the binary images (Passalacqua et al., 2013). Clusters (usually 2-3 pixels in size) that appear inside the river network do not correspond to bars or islands. We found frequent areas where strong reflection from the bed sediment cause water pixels to appear more like sand. Isolated water pixels that do not belong to the river are located in water-logged areas. We identify these types of errors and reprocess the binary images to remove them automatically.

For this, we first identify the isolated water patches from the binary images. To do this, we define a search window of $7 \times 7$ pixel size. We run this window on the image and look for neighboring water pixels in all surrounding directions. If a water pixel in the classified image is disconnected in all directions from the neighboring water pixels for more than seven pixels, we consider them as isolated water bodies. We therefore re-classify such pixels as dry. We re-iterate this procedure by applying a region growing algorithm (Mehnert and Jackway, 1997; Bernander et al., 2013; Fan et al., 2005). For this we initially select a water

seed pixel inside the river channel. The algorithm uses the initial water pixels and starts growing. This procedure removes all isolated water patches from the binary image, and retains only water pixels connected to the river network.

Once images are reclassified, we reprocess them to merge the water pixels that were initially classified as dry inside a river channel. For this we define a search window of $3 \times 3$ pixels. We choose this size by assuming that dry pixels should be more than 90 meter in size to be considered as bars or islands. Otherwise, such pixels are treated as water pixels. We move the search

window on the binary image and look for neighboring dry pixels inside the river channel.

Similarly, to identify river pixels from Sentinel 1A images, we use VH (Vertical transmission and Horizontal reception) polarized band. We have Sentinel Application Platform (SNAP) v6.0 to perform the radiometric calibration, speckle noise reduction using refined Lee filter and terrain corrections and finally generate the backscatter ($\sigma_0$) image. This image has pixel size of $60 \times 60m$ that we resampled at $30 \times 30m$ to be consistent with the pixel size of Landsat images. In microwave region,

open and calm water bodies exhibit low backscatter values due to high specular reflection from the water surface (Schlaffer et al., 2015; Twele et al., 2016; Amitrano et al., 2018). We manually set a threshold value to separate water and dry pixels from

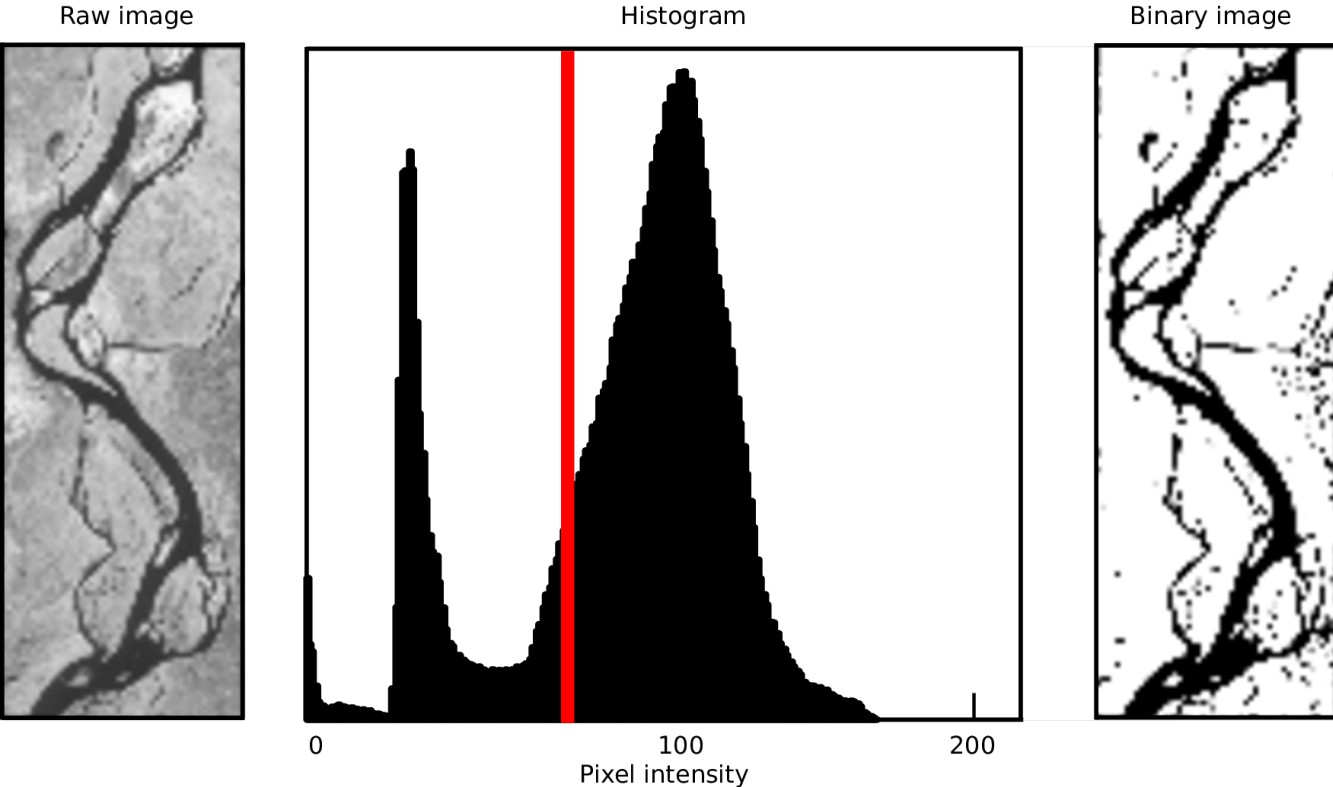

| Raw image | Histogram | Binary image |

**Figure 4.** Histogram showing the distribution of pixel gray level intensity values. The optimal threshold (T) value (marked with red line) is obtained from the iterative threshold selection algorithm.

Sentinel-1 images. Finally, we follow a similar procedure as we developed for Landsat images to process the binary image obtained from Sentinel-1.

Once the satellite images are classified, we use the binary images to extract the width of each channel. We do this by measuring the distance from the center of a channel to its banks orthogonally to the flow direction. A detailed automated procedure of width extraction of a river channel is given in Appendix B.

## 5   Result

### 5.1   Accuracy assessment

To assess the precision with which we can estimate the discharge of a thread, we need to quantify the accuracy of our width-
200 extraction procedure using Landsat and Sentinel-1 satellite images. To evaluate this, we superimpose the contours of river channels, extracted using our algorithm, to the original gray-scale images used for the extraction. We then carefully check for a match between the contours boundary and water boundary in gray scale image. We observed a good agreement between

automatically extracted channel boundary and the edge of the water line in gray image. However, our algorithm fails to extract

the contours of the smallest channels (60 - 90 meter in width). Several reasons explain this limitation. First, as these channels are

both shallow and only a few pixels wide, their pixel intensity is close to the pixel intensity of dry areas. Therefore, the optimal

threshold applied to categorize the image pixels does not identify these channels as water. Second, although an increase in

the classification threshold could force the algorithm to identify these pixels as water, it would also add significant noise by

classifying many dry pixels as water pixels. Such a limitation appears to be closely related to the image resolution.

Given this qualitative agreement, we proceed to evaluate the accuracy of the width extraction procedure. To do this, we

overlay the transects used by the algorithm to measure the width of a thread on the original image (Fig. 5 a). We then manually

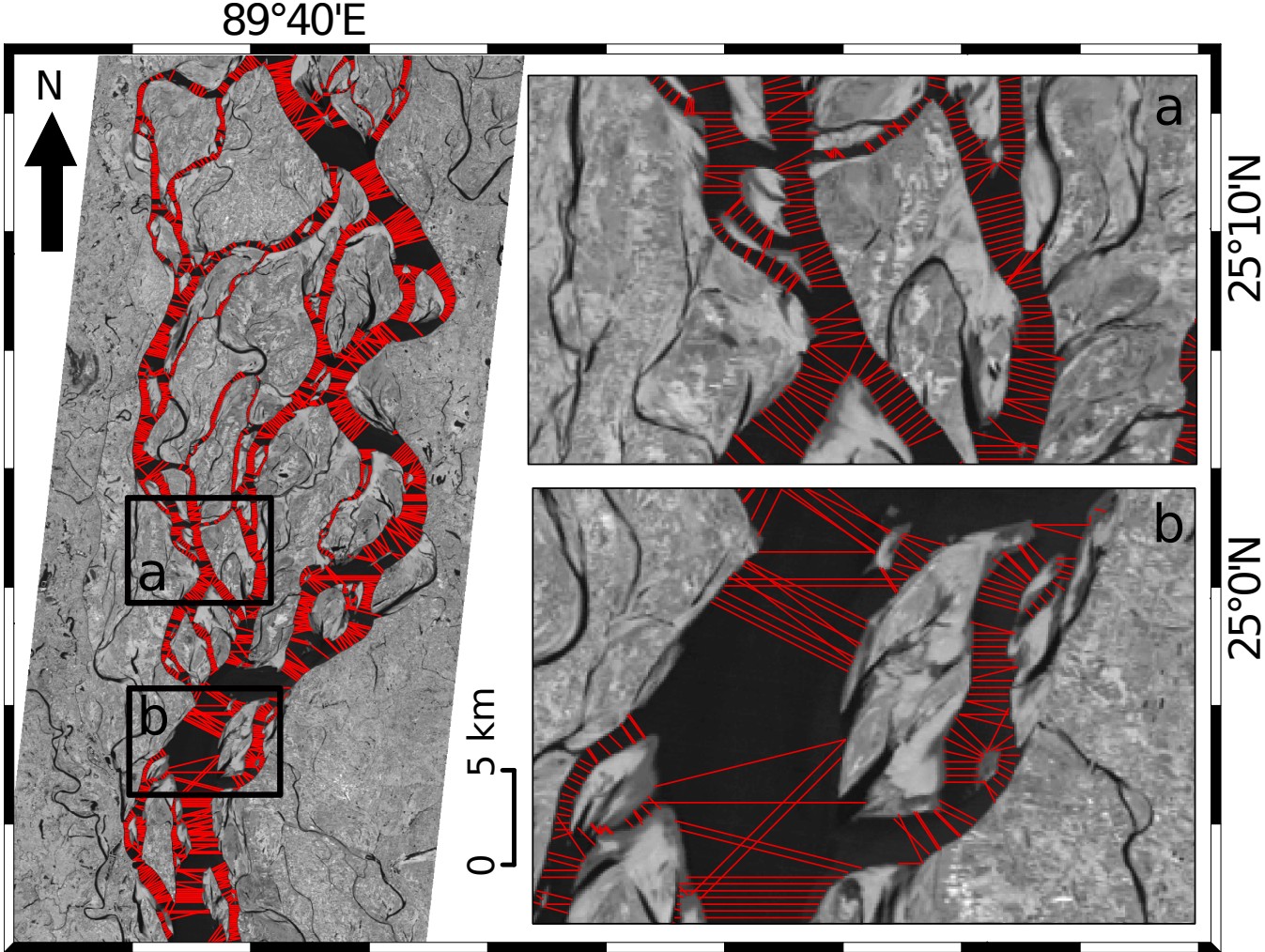

**Figure 5.** Width of the individual threads estimated across different transects along a reach of the Brahmaputra river from Landsat satellite image. Windows (a) and (b) illustrate the regions of valid and erroneous transects at different places in the river (image source: Landsat-TM, 29 November 2013).

measure the width at randomly selected transects for comparison. For each river, we manually measure the width at more than 15 randomly selected transects. We then compare the automatically extracted and manually measured widths.

Figure 6 compares the widths extracted automatically and manually. Most of the data points cluster on the 1:1 line. This indicates that, for the vast majority of threads, the width computed from our automated procedure is almost equal to the width measured manually.

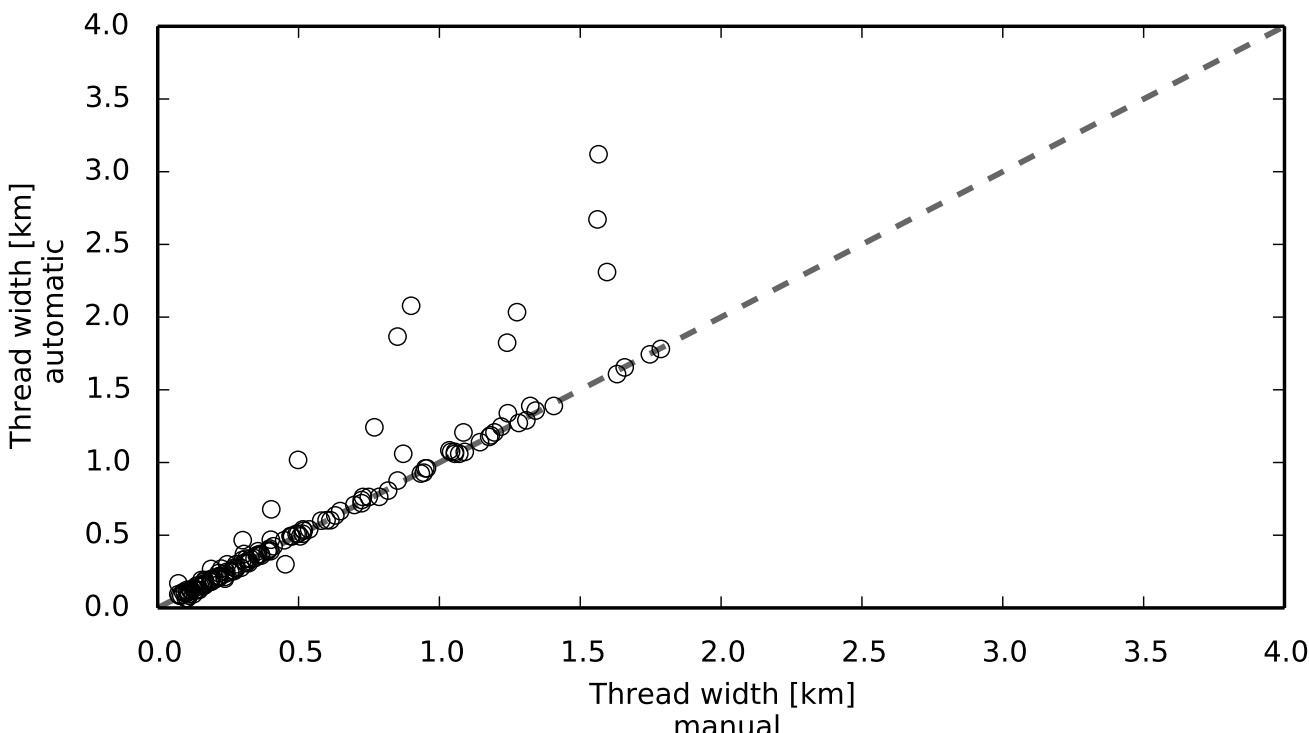

**Figure 6.** Threads width extracted using automated technique is plotted as function of width extracted manually.

There are some outliers however. They correspond to places along the threads where our automated procedure draws erroneous transects (Fig. 5 b). Most of such transects are located near highly curved reaches at the confluence or diffluence of two or more threads. In such places, the width of a thread is overestimated sometimes by more than $50\%$ compared to the width measured manually. At most locations though, our procedure extracts valid transects (Fig. 5 a).

Further, we assess the distribution of relative discrepancies between automatically and manually measured widths (Fig. 7).
To quantify the precision of our measurement we compute the relative error. We observe that the relative error of $90\%$ of our measurements is centered around a mean $\mu \approx -0.02$ with a standard deviation $\sigma \approx 0.07$. This validates the width-extraction procedure.

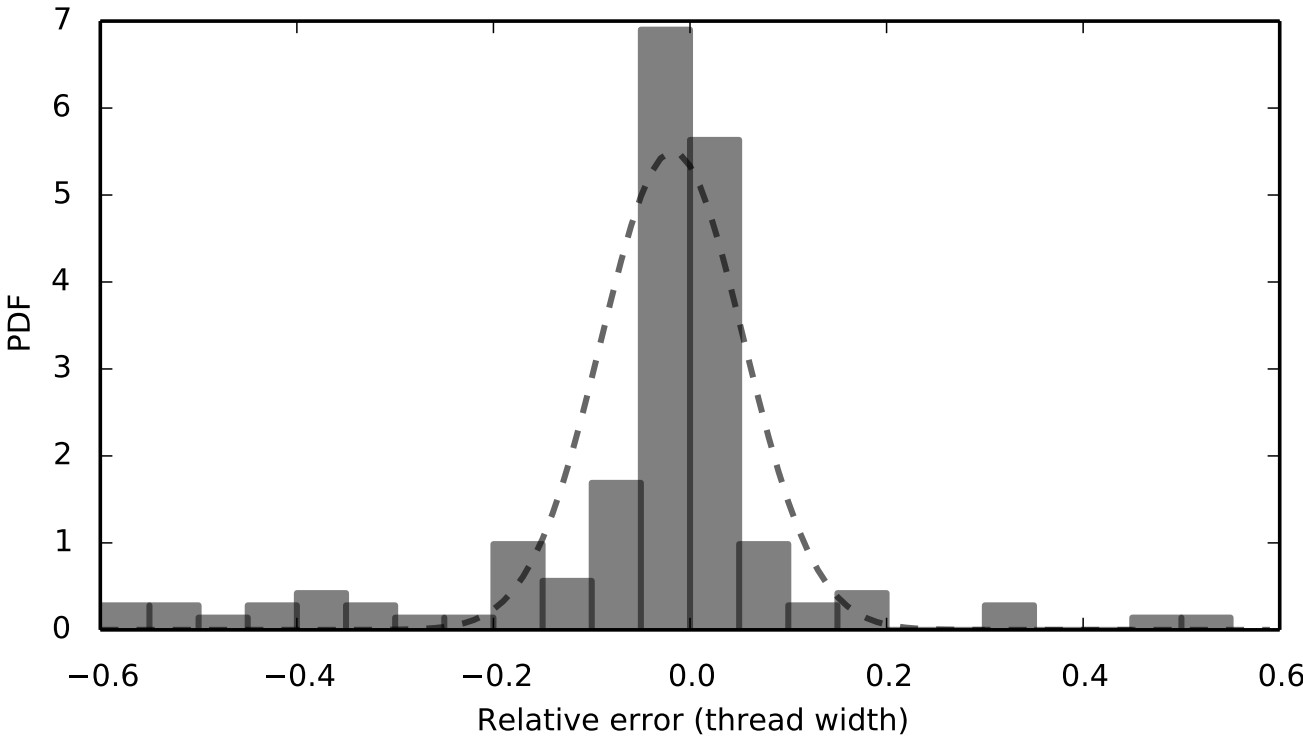

**Figure 7.** Distribution of the error in the threads width extracted automatically. The corresponding normal distribution is obtained by removing the $10\%$ extreme values from the distribution.

## 5.2 Width variability along a thread

Particularly in a braided river, the width of a thread varies significantly along its course. To quantify this variability, we select a reach and plot the probability distribution of the width measured across different transects. We observed that the distribution of width histograms is skewed Figure 8. This skewness results from the natural variability of width along the course and also due to the error in width extraction from images, particularly at the location where the curvature of a thread is high. The resulting skewness will be amplified in the discharge histogram because of the non-linear relationship relating the two variables. To take the skewness into account, we have calculated the geometric mean of all the measured values. The geometric mean is less affected by extreme values in a skewed distribution and can be considered a representative width ($W_r$). However, in meandering rivers where the variability in width within a reach is not much, arithmetic mean can be considered a representative width.

## 5.3 Discharge estimation

We now proceed to estimate discharge ($Q_w$) for the Himalayan Rivers based on their channel widths extracted from satellite images. To have a meaningful comparison between the image derived discharge and the corresponding in-situ measurement,

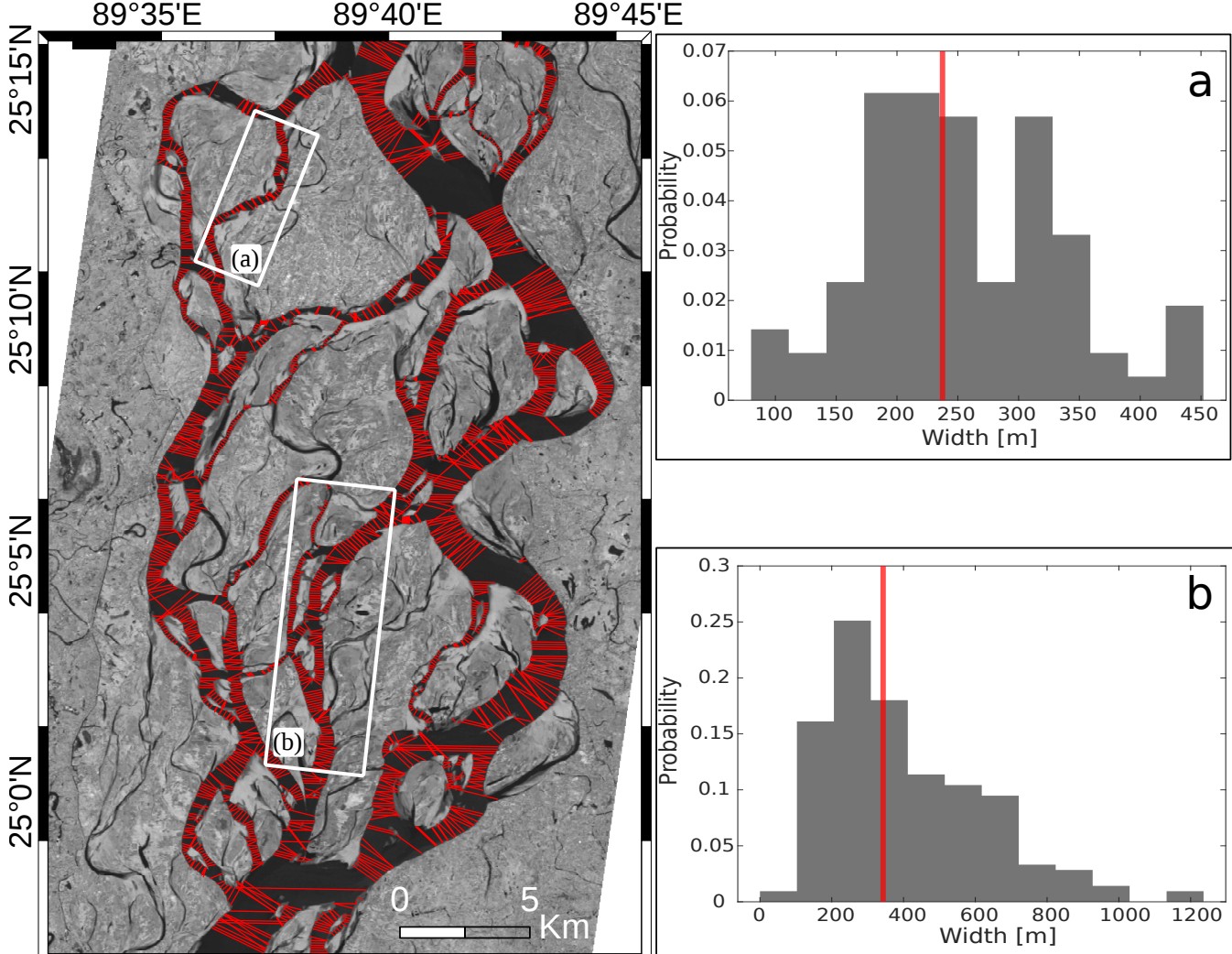

**Figure 8.** Spatial distribution of the width measured along threads of a braided reach of the Ganga River near the Paksay Gauge station in Bangladesh. Vertical line (red) on the histograms (a and b) is the geometric mean that corresponds to the most probable width (Wm).

we select a reach about ten times longer than the width of a river on satellite images. In the case of a braided river, we consider the widest channel to define reach length. In the selected reach we assume that discharge is conserved, there is no significant addition or extraction of water in the river.

To estimate discharge of the study reach, we use a regime relation established by Gaurav et al. (2017) based on threshold channel theory (Eq. 4) and field measurements of channel's width and discharge on the Ganga-Brahmaputra Plain. The resulting regime relation is governed by:

$$Q_w = \left(\frac{W_r}{\alpha}\right)^2 \sqrt{(gd_s)}, \tag{6}$$

where $\alpha$ is the best-fit coefficient, an empirical value obtained from fitting the prefactor of the regime curve (Eq.4), $W_r$ is the representative width and $d_s$ is the median grain size.

We use Eq. 6 to calculate the discharge for threads of known width. Because the river width scales non-linearly with discharge, regime relations obtained refer to the total width in the case of a braided river; and will not be the same as those obtained for individual threads. Since most of the studied rivers are braided, we first calculate the discharge for individual threads across a given section. We then sum the discharge of the individual threads across a transect to compute the total discharge at a section.

### 5.4   Estimated Vs. measured discharge

Once monthly discharges for all the rivers are estimated from satellite images, we compare them with the average monthly discharge measured at the corresponding gauge stations. To do so, we plot the hydrographs of the estimated and measured discharges together (Fig. 9). We observed that the estimated discharge of the Kosi, Ganga, and Brahmaputra rivers from satellite images is overestimated and almost constant throughout during the non-monsoon period (October-May). Conversely, the Indus, Chenab, and Teesta rivers show a clear annual cycle. This observed trend is not entirely clear to us, it could possibly be related to the flow regulation, as these river are highly regulated through a series of dams and barrages.

Further we observed that the estimated discharge for most of our rivers show a significant rise during the monsoon period (June-September). To the first order, it appears that our approach is able to capture the rising trend of discharges during the monsoon period, however the estimated discharges are lesser than the measured discharges. Table 2 compares the estimated and measured discharges during the monsoon period. For most of our rivers, the difference between measured and estimated discharges is less than $50\%$; though this difference is comparatively high for the Indus ($72 - 78\%$) and Chenab ($36 - 67\%$) rivers (Table 2).

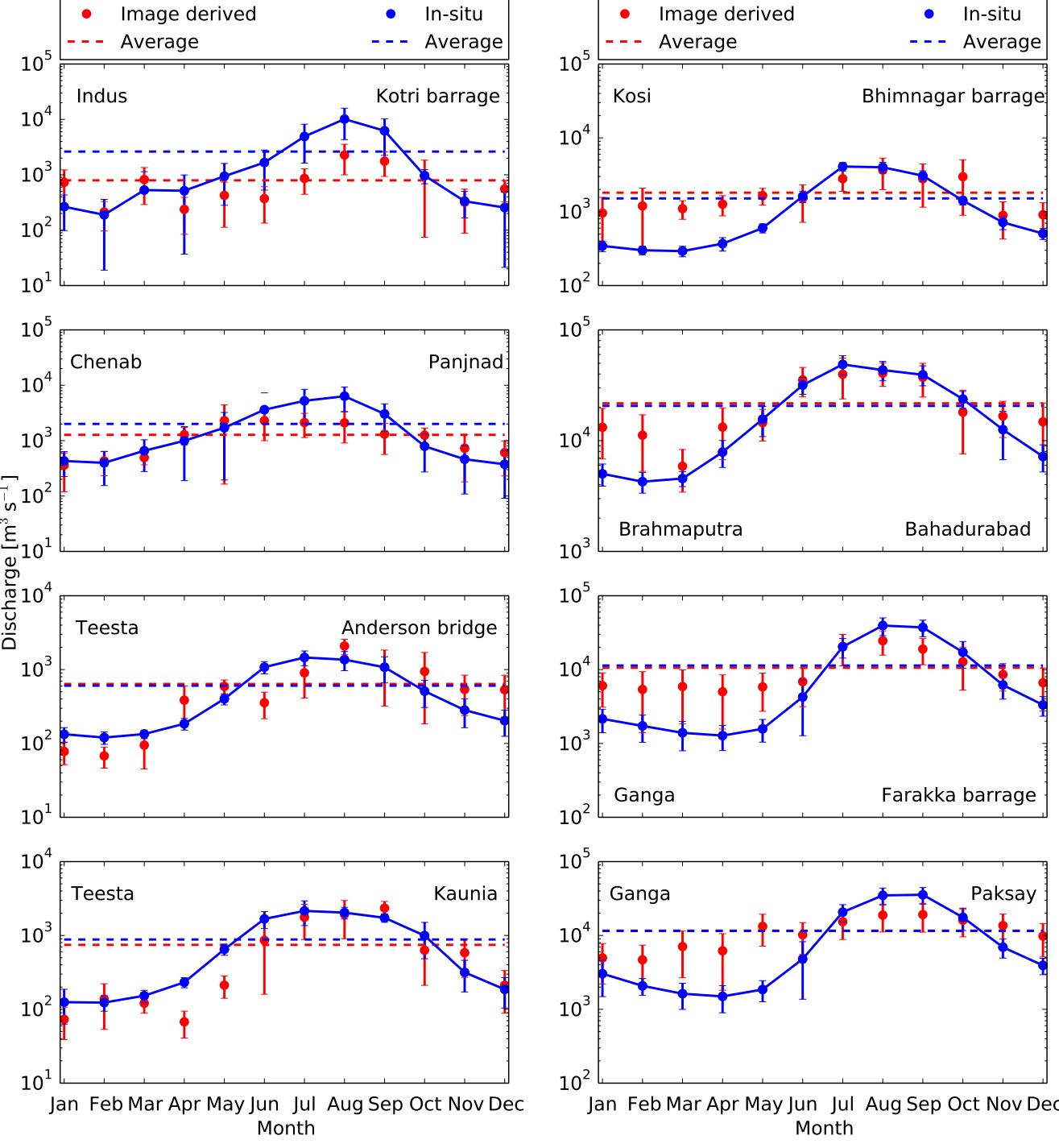

**Figure 9.** Hydrograph of satellite derived reach averaged discharge against their monthly average discharge recorded at the gauging station. Error bar in measured discharge (blue) is the standard deviation calculated from the time series of different month. Error bar in estimated discharge represents the standard deviation within the study reach. Dotted red and blue lines are the annual average discharge obtained from satellite images and in-situ measurement respectively.

| Rivers | | Jun | Jul | Aug | Sept |
|---|---|---|---|---|---|
| | | \multicolumn{4}{c}{Monsoon discharge $[m^3 s^{-1}]$} | | | |

| | Monsoon discharge $[m^3s^{-1}]$ | | | |
|---|---|---|---|---|

| Rivers | | Jun | Jul | Aug | Sept |
|---|---|---|---|---|---|
| Kosi (Bhimnagar barrage) | In-situ | $1616 \pm 285$ | $4091 \pm 530$ | $3998 \pm 660$ | $3072 \pm 509$ |
| | Image derived | $1515 \pm 797$ | $2800 \pm 912$ | $3660 \pm 1667$ | $2796 \pm 1644$ |
| | Difference (%) | -6 | -32 | -8 | -9 |
| Brahmaputra (Bahadurabad) | In-situ | $31717 \pm 5536$ | $48769 \pm 9640$ | $43387 \pm 8722$ | $39320 \pm 8071$ |
| | Image derived | $35335 \pm 10491$ | $39716 \pm 15914$ | $40653 \pm 9808$ | $37316 \pm 12580$ |
| | Difference (%) | 11 | -19 | -6 | -5 |
| Ganga (Farakka barrage) | In-situ | $4260 \pm 2989$ | $20375 \pm 6059$ | $39462 \pm 10665$ | $37264 \pm 9415$ |
| | Image derived | $6864 \pm 3717$ | $20599 \pm 9343$ | $24562 \pm 8871$ | $18971 \pm 7364$ |
| | Difference (%) | 61 | 1 | -38 | -49 |
| Ganga (Paksay) | In-situ | $4794 \pm 3425$ | $20691 \pm 5427$ | $34887 \pm 9002$ | $35546 \pm 8985$ |
| | Image derived | $10226 \pm 4689$ | $15333 \pm 6510$ | $18862 \pm 7691$ | $19168 \pm 8089$ |
| | Difference | 113 | -26 | -46 | -46 |
| Teesta (Anderson bridge) | In-situ | $1078 \pm 204$ | $1458 \pm 330$ | $1363 \pm 395$ | $1076 \pm 416$ |
| | Image derived | $356 \pm 139$ | $904 \pm 494$ | $2086 \pm 494$ | $1079 \pm 759$ |
| | Difference (%) | -67 | -38 | 53 | 0 |
| Teesta (Kaunia) | In-situ | $1674 \pm 428$ | $2151 \pm 792$ | $2037 \pm 369$ | $1733 \pm 227$ |
| | Image derived | $860 \pm 700$ | $1765 \pm 883$ | $1938 \pm 1036$ | $2346 \pm 540$ |
| | Difference (%) | -49 | -18 | -5 | 35 |
| Indus (Kotri barrage) | In-situ | $1665 \pm 1136$ | $4912 \pm 3290$ | $10128 \pm 5807$ | $6227 \pm 3980$ |
| | Image derived | $372 \pm 238$ | $861 \pm 417$ | $2279 \pm 1279$ | $1759 \pm 826$ |
| | Difference (%) | -78 | -82 | -77 | -72 |
| Chenab (Panjnad) | In-situ | $3621 \pm 2812$ | $5235 \pm 3206$ | $6340 \pm 2983$ | $3038 \pm 1574$ |
| | Image derived | $2300 \pm 1302$ | $2125 \pm 988$ | $2099 \pm 1193$ | $1311 \pm 749$ |
| | Difference (%) | -36 | -59 | -67 | -57 |

**Table 1.** Comparison between the image derived and in-situ measured discharge of the Himalayan rivers during the Indian summer monsoon period. Error in measured discharge is the standard deviation calculated from the time series of different month. Error in estimated discharge is the standard deviation within the study reach.

## 6  Discussion

It is important to note that the discharge estimated from satellite images does not correspond to an instantaneous discharge. To understand the emergence of constant hydrograph from the estimated discharge derived from satellite images we explore the concept of channel forming (formative) discharge i.e. a discharge that sets the geometry of alluvial river channels. Several

workers Inglis and Lacey (1947); Leopold and Maddock (1953); Blench (1957), have shown that the geometry of an alluvial channel corresponds to a formative discharge (see table A3 in appendix for the definition of different discharges). They have discussed how a limited range of flows are responsible for shaping its channel. At low-flow discharge, the water simply flows through the threads without affecting their geometry. Schumm and Lichty (1965) used the concept of time span (geologic, modern and present) in defining the interrelationship between dependent and independent variables of a river system. According

to them, morphology of a river channel is set in the modern time span (last 1000 years) by the average discharge of water and sediment. In the present time span (1 year or less), channel morphology can be considered as independent variable against instantaneous discharge of water and sediment.

Similarly, it has been argued by Inglis and Lacey (1947); Leopold and Maddock (1953); Blench (1957) that it is not the highest flows that contribute the most in shaping a river channel. Such high discharges are capable of transforming the channel,

but they occur so infrequently that, on average, their morphological impact is small. Wolman and Miller (1960) highlighted that the bankfull discharge that occurs once each year or every two years sets the pattern and channel width of the alluvial rivers. Formative discharge for the Himalayan rivers is expected to occur in the monsoon period, thus one may expect that during low flow such rivers maintain their flows without modifying the existing channel geometry (Roy and Sinha, 2014). This clearly reflects in the discharge hydrographs estimated from the measurement of channel's width from the satellite images (Fig. 9).

Furthermore, Métivier et al. (2017) have recently shown that non cohesive streams laden with sediments cannot have a width much larger than the width of a threshold stream before they start to braid. They also showed that, for experimental braided rivers, threads are always formed at the bankfull flow, and at the limit of stability. Our hypothesis is thus that the formative discharge of threads in the Ganga plain is the bankfull discharge. This is probably why for most of the rivers our estimated discharge from satellite images remain constant throughout during the non-monsoon period and is mostly overestimated than

the measured discharge at gauge stations.

According to Inglis and Lacey (1947), rivers approach their equilibrium geometry for a formative discharge that approximately corresponds to the bankfull discharge. They suggested this discharge lies between 1/2 and 2/3 of the maximum discharge. It has also been suggested that the formative discharge corresponds to the median discharge (Blench, 1957). In their study, Leopold and Maddock (1953) used the discharge that corresponds to a given frequency of occurrence and compared it to

the hydraulic geometry of the river. Based on their observations in the United States, they recommended the use of the annual average discharge as a proxy for the formative discharge. Hereafter, we use the definition of Leopold and Maddock (1953).

Based on our understanding of the geometry of alluvial river channels, we argue that the width of the thread that we extract from satellite images corresponds to a formative discharge. As discussed by Gaurav et al. (2014), high-resolution bathymetry profile of a braided thread reveals complex topography of the bed. For an example, Figure C1 in appendix, illustrates the

cross-section of a braided thread measured in the field using Acoustic Doppler Current Profiler (ADCP). This high resolution bathymetric profile enables us to identify the two different threads separated by submerged bars and islands. In such situations, discharge is not carried across the width seen from the plan view, but only through the narrow active regions. Further, this indicates that during the low flow water spread in the existing geometry is set at the formative discharge. Currently, satellite images allow us to measure the top width of the water surface. We presume for a given thread, discharge estimated from these widths should compare with the formative discharge.

We now evaluate how the discharge estimated from satellite data varies with time. We plot the monthly discharge estimated for all of our rivers to their corresponding average monthly discharge measured at the gauge stations (Fig. 9). The monthly average discharge of the Himalayan Foreland rivers appears to be a representative of the actual hydrograph (Fig. C2). As suggested earlier by Inglis and Lacey (1947); Leopold and Maddock (1953) and Blench (1957) we observe that except for the Indus, Chenab, and Teesta river, the estimated discharges from images are nearly constant throughout during the non-monsoon period, with small fluctuations around their mean. This supports the hypothesis that the width of the thread extracted from satellite images corresponds to a value closer to the formative discharge.

To go further, now we compare the annual average discharge estimated from Landsat and Sentinel-1A images for different months to the annual average discharge measured at corresponding ground stations. We plot these discharges on a log-log scale (Fig. 10). The discharge estimated from satellite images agrees to an order of magnitude with the measured discharge.

The difference between measured and estimated annual average discharges for the Brahmaputra, Ganga, Kosi, and Teesta rivers is less than $20\%$. However, this difference is comparatively high for the Indus ($78\%$) and Chenab ($49\%$) rivers. Interestingly, the estimated discharge for the Teesta (at Anderson station), Ganga (at Farakka & Paksay) and Brahmaputra (at Bahadurabad) rivers converge to their measured discharge with a small difference of $5\%$, $8\%$, $4\%$ and $<1\%$, respectively (table 2); whereas the estimated discharge of the Teesta (at Kaunia station) and Kosi (at Bhimnagar) show a relatively higher difference of $19\%$, and $16\%$. This difference could be possibly related to the anthropogenic impact on the natural flow condition. For example, the selected study reaches for the Teesta (at Kaunia station) and Kosi (at Bhimnagar) rivers is located near the barrage where flow is regulated. However, this relationship is not entirely clear at this stage.

Similarly, the observed annual cycle in discharge and large difference between the estimated and measured discharge of the Indus and Chenab rivers could possibly be related to a series of dams and barrages (Kotri barrage, 1955, Mangla dam, 1967, Tarbela dam, 1976) that have been constructed. Such interventions have significantly altered the water and sediment discharge of the Indus river. For example, downstream of the Kotri barrage, the average annual water discharge of the Indus river has declined at an alarming rate of about (107 to 10) $\times 10^9 \, \mathrm{m}^3$ from 1954 to 2003 (Inam et al., 2007). This continuous decline in the average annual discharge might have significantly modified the geometry of the Indus river.

Further to understand our estimates of discharge for the Chenab, Indus and Teesta rivers, we plot their monthly discharge time series recorded at the corresponding gauge station together with the discharge estimated from satellite images (Fig. C3 and C4). Despite a large variability, the discharge time series of Indus and Chenab rivers show a strong declining trend during the monsoon period (June-September); whereas discharge during the non-monsoon period appears to remain constant around the mean value. Figure (C3) clearly show that discharge estimated from satellite images plot within the variability of the observed

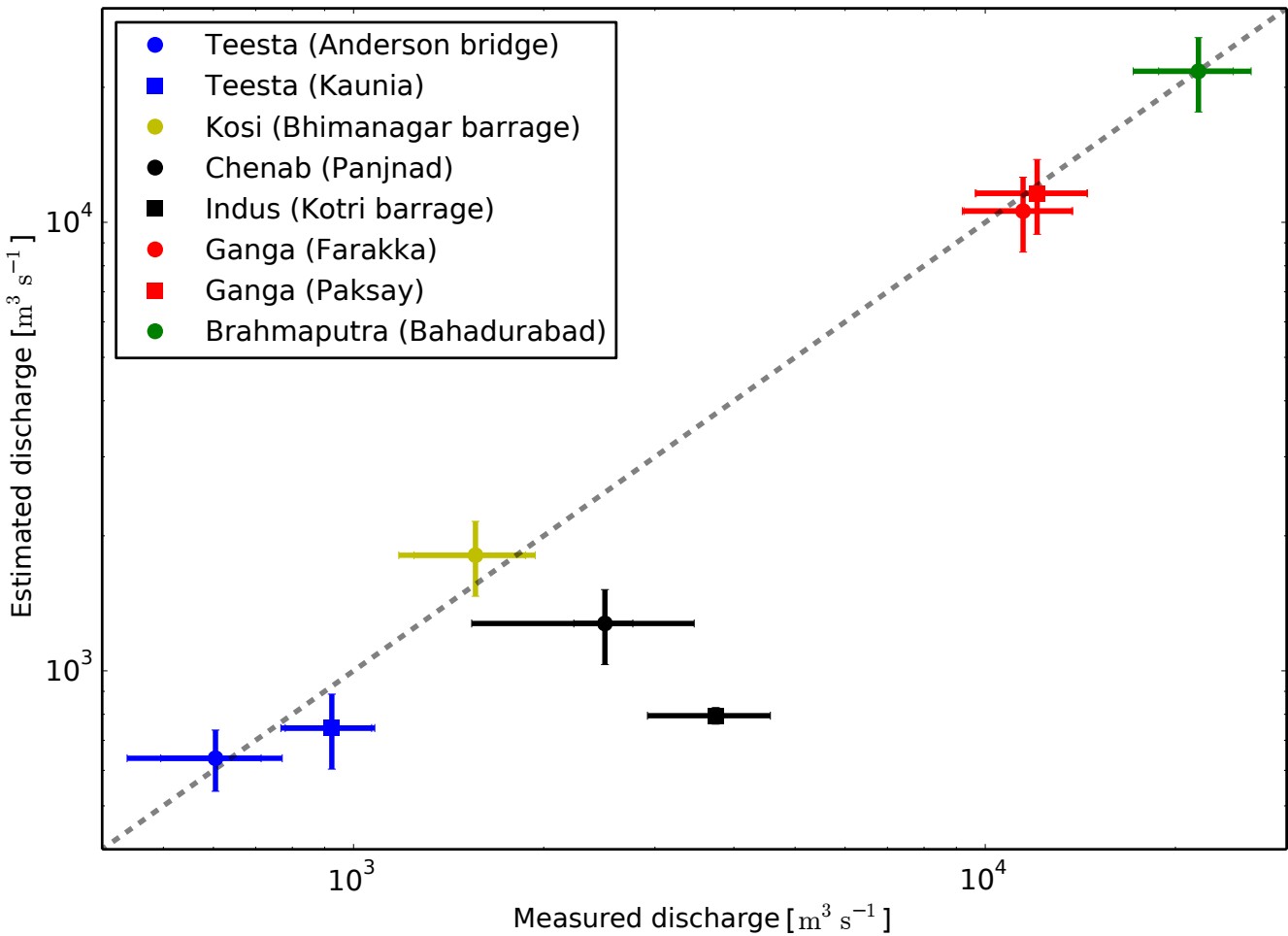

**Figure 10.** Satellite-derived river discharge against annual average discharge measured at a ground station. Error bar in measured discharge represent the standard deviation. Error bar for estimated discharge is calculated by considering ±10% measurement uncertainty in the channels width from satellite images.

trend. The estimated discharge of the Teesta River also plot within the noise of the observed trend. This gives us confidence in our estimates of discharge, especially for the rivers for which we have a limited and old record (1973-1979) of in-situ discharges.

In a recent study Allen and Pavelsky (2018) measured the width of the global rivers from Landsat images for the month when they commonly flow near mean discharge. We have used the water mask binary images from "Global River Width from Landsat (GRWL) database" and measure the threads width of the Brahmaputra, Chenab, Ganga, Indus and Teesta rivers (Appendix C). We used these widths to estimate discharges using our regime curve and compare them with the mean annual discharge recorded at the corresponding gauge station and our estimates from satellite images. We observed for most of our

| River | Station | $\langle Q_{insitu} \rangle$ $m^3 s^{-1}$ | $\langle Q_{sat.} \rangle$ $m^3 s^{-1}$ | $Q_{diff.}$ $m^3 s^{-1}$ | $Q_{diff.}$ % |
|-------|---------|------|------|------|------|
| Teesta | Anderson | $605 \pm 109$ | $638 \pm 165$ | 33 | 5 |
| Teesta | Kaunia | $924 \pm 144$ | $745 \pm 155$ | -179 | 19 |
| Kosi | Bhimnagar | $1559 \pm 313$ | $1810 \pm 380$ | 251 | 16 |
| Chenab | Panjnad | $2500 \pm 961$ | $1275 \pm 268$ | -1225 | 49 |
| Indus | Kotri | $3745 \pm 825$ | $794 \pm 162$ | -2951 | 78 |
| Ganga | Farakka | $11477 \pm 2279$ | $10593 \pm 2225$ | -884 | 8 |
| Ganga | Paksay | $12080 \pm 2403$ | $11605 \pm 2438$ | -475 | 4 |
| Brahmaputra | Bahadurabad | $21751 \pm 2942$ | $21717 \pm 4740$ | -34 | < 1 |

**Table 2.** Comparison between the annual average discharge measured at the gauge stations and estimated from satellite images.

rivers the discharge estimated from thread's width extracted from the GRWL database of Allen and Pavelsky (2018) falls within the same order of magnitude to the yearly average discharge measured at the corresponding gauge stations (Table C1). For most
rivers, the difference between measured and estimated discharge from GRWL data are approximately less than 60%. However this difference is comparatively high for the Kosi (88%) and Indus (95%) river. Interestingly, in accordance to our estimates, GRWL database also show a high reduction in discharge of the Indus and Chenab river from the measured discharge at their corresponding ground stations during the period 1973-1979.

The GRWL database is a first ever compilation of width for the global rivers, it may be used together with our regime curve
to obtain a first order approximation of formative discharge of the Himalayan Foreland rivers.

## 7 Conclusion and future outlook

The semi-empirical regime relation established by Gaurav et al. (2017) and remote sensing images can be used to obtain a first order estimate of the formative discharge of the Himalayan Foreland, if their channel width is known. The regime equation used here is established from the recent measurements and the published data by Gaurav et al. (2014, 2017). This
equation is based on threshold theory and instantaneous measurements of hydraulic geometry of individual threads of various braided and meandering rivers. The measurements were acquired during the period when rivers of the Himalayan Foreland usually flow at their formative discharge (Roy and Sinha, 2014). Therefore, this regime equation only provides an estimate of the formative discharge, and it can not capture the instantaneous variations of discharge. On the other hand, as this regime relation is established from the measurements in braided and meandering rivers, it can be used to estimate first order estimate
for formative discharge in a river of any planform. It is especially useful and relevant for braided rivers that present several difficulties for the measurement of discharge in the field. Our regime equation requires only one parameter (grain-size) to estimate discharge from width measurements. It can be obtained easily from field measurements. Since our regime equation is established from measurement of a wide range of channels spanning over the Ganga and Brahmaputra plains, we believe

it can be used to obtain the first order estimate of formative discharge of rivers in the Himalayan Foreland by just measuring their channel width on satellite or aerial images. Using our semi-empirical regime equation and satellite images of Landsat and sentinel-1 missions, we have estimated the discharge of six major rivers in the Himalayan Foreland (Brahmaputra, Chenab, Ganga, Indus, Kosi, and Teesta). Our estimated discharges closely compare with the average annual discharge measured at the nearest gauging stations. This first-order agreement although encouraging, requires further research to improve the degree of agreement between measured and estimated discharges. One of the main sources of uncertainty in discharge estimate is due to the error in the measurement of thread's width. This depends on the image resolution and accuracy of the algorithm used for extraction of river pixels from remote sensing images. A better resolution remote sensing images would most likely minimise the uncertainty and improve the agreement between estimated and in-situ discharge. Further our regime equation established for the Himalayan rivers is based on a simple physical mechanism that explains the geometry of alluvial channels. We therefore suspect that the procedure we have established could be extended to most alluvial rivers. Globally it has been observed that the threshold theory well predicts the exponent of the regime equation (Eq. 6), however the prefactor may vary significantly depending on the grain size distribution, turbulent friction coefficient and the critical Shield parameter (Métivier et al., 2017). It is therefore suggested to modify this regime curve from the measurement of width, discharge and grain size of a individual threads of alluvial channels in the field before applying it to the rivers of different climatic regime. Further, it should be noted that our regime curve relates to the measurement of hydraulic geometry of individual threads of braided and meandering rivers, therefore it is applicable only at the thread scale. Since the resulting regime curve is non linear, estimating discharge across a transect in a braided river from the aggregated width will be different from the one obtained after the summation of discharges of the individual threads.

This study presents a robust methodology and is a step towards obtaining first order estimates of formative discharge in ungauged river basins solely from remote sensing images. It can be used for sustainable river development and management to ensure regional water security and flood management, especially in the regions where river discharge data is not readily available.

## Appendix A: Dataset

### A1   Satellite images

Detailed specification of satellite data (Landsat and Sentinel-1) used in this study. Table A1

| Landsat-TM & Landsat-8 | | | | |
|------------|------------|----------------------|-----------|---------------|
| **River** | **Date** | **Scene ID** | **Satellite** | **Gauge station** |
| Brahmaputra | 2009-01-18 | LT51380432009018KHC01 | L-TM | 25.18/89.66 |
| Brahmaputra | 2009-02-19 | LT51380432009050KHC00 | L-TM | 25.18/89.66 |
| Brahmaputra | 2014-03-21 | LC81380432014080LGN00 | L-8 | 25.18/89.66 |
| Brahmaputra | 2014-04-22 | LC81380432014112LGN00 | L-8 | 25.18/89.66 |

| | | | | |
|---|---|---|---|---|
| Brahmaputra | 2014-10-31 | LC81380432014304LGN00 | L-8 | 25.18/89.66 |
| Brahmaputra | 2013-11-29 | LC81380432013333LGN00 | L-8 | 25.18/89.66 |
| Brahmaputra | 2014-12-02 | LC81380432014336LGN00 | L-8 | 25.18/89.66 |
| Chenab | 2014-01-04 | LC81500402014004LGN00 | L-8 | 29.35/71.30 |
| Chenab | 2018-02-16 | LC81500402018047LGN00 | L-8 | 29.35/71.30 |
| Chenab | 2015-04-13 | LC81500402015103LGN00 | L-8 | 29.35/71.30 |
| Chenab | 2014-05-28 | LC81500402014148LGN00 | L-8 | 29.35/71.30 |
| Chenab | 2014-06-29 | LC81500402014180LGN00 | L-8 | 29.35/71.30 |
| Chenab | 2014-07-15 | LC81500402014196LGN00 | L-8 | 29.35/71.30 |
| Chenab | 2014-10-19 | LC81500402014292LGN00 | L-8 | 29.35/71.30 |
| Chenab | 2013-11-01 | LC81500402013305LGN00 | L-8 | 29.35/71.30 |
| Chenab | 2014-12-06 | LC81500402014340LGN00 | L-8 | 29.35/71.30 |
| Ganga | 2015-02-11 | LC81390432015042LGN01 | L-8 | 24.83/87.92 |
| Ganga | 2015-03-15 | LC81390432015074LGN00 | L-8 | 24.83/87.92 |
| Ganga | 2013-04-02 | LT51390432010092KHC00 | L-8 | 24.83/87.92 |
| Ganga | 2014-06-16 | LC81390432014167LGN00 | L-8 | 24.83/87.92 |
| Ganga | 2014-11-23 | LC81390432014327LGN00 | L-8 | 24.83/87.92 |
| Ganga | 2009-01-18 | LT51380432009018KHC01 | L-TM | 24.08/89.03 |
| Ganga | 2009-02-19 | LT51380432009050KHC00 | L-TM | 24.08/89.03 |
| Ganga | 2014-03-21 | LC81380432014080LGN00 | L-8 | 24.08/89.03 |
| Ganga | 2014-04-22 | LC81380432014112LGN00 | L-8 | 24.08/89.03 |
| Ganga | 2014-10-31 | LC81380432014304LGN00 | L-8 | 24.08/89.03 |
| Ganga | 2013-11-29 | LC81380432013333LGN00 | L-8 | 24.08/89.03 |
| Ganga | 2014-12-02 | LC81380432014336LGN00 | L-8 | 24.08/89.03 |
| Indus | 2015-01-05 | LC81520422015005LGN00 | L-8 | 25.35/68.35 |
| Indus | 2017-02-11 | LC81520422017042LGN00 | L-8 | 25.35/68.35 |
| Indus | 2015-03-10 | LC81520422015069LGN00 | L-8 | 25.35/68.35 |
| Indus | 2014-04-24 | LC81520422014114LGN00 | L-8 | 25.35/68.35 |
| Indus | 2014-06-27 | LC81520422014178LGN00 | L-8 | 25.35/68.35 |
| Indus | 2014-10-17 | LC81520422014290LGN00 | L-8 | 25.35/68.35 |
| Indus | 2014-11-18 | LC81520422014162LGN00 | L-8 | 25.35/68.35 |
| Kosi | 1991-01-15 | LT51400421991015ISP00 | L-TM | 26.52/86.92 |
| Kosi | 2011-02-07 | LT51400422011038BKT00 | L-TM | 26.52/86.92 |
| Kosi | 1992-03-06 | LT51400421992066ISP00 | L-TM | 26.52/86.92 |
| Kosi | 2018-05-01 | LC81400422018121LGN00 | L-8 | 26.52/86.92 |

| Kosi | 2015-09-30 | LC81400422015273LGN01 | L-8 | 26.52/86.92 |
|------|------------|------------------------|-----|-------------|
| Kosi | 2000-10-14 | LE71400422000288SGS00 | L-TM | 26.52/86.92 |
| Kosi | 2013-11-11 | LC81400422013315LGN00 | L-TM | 26.52/86.92 |
| Kosi | 2002-12-07 | LE71400422002341SGS00 | L-TM | 26.52/86.92 |
| Teesta | 2014-04-22 | LC81380422014112LGN00 | L-8 | 26.33/88.87 |
| Teesta | 2014-10-31 | LC81380422014304LGN00 | L-8 | 26.33/88.87 |
| Teesta | 2014-11-16 | LC81380422014320LGN00 | L-8 | 26.33/88.87 |
| Teesta | 2014-12-02 | LC81380422014336LGN00 | L-8 | 26.33/88.87 |
| Teesta | 2015-03-08 | LC81380422015067LGN00 | L-8 | 25.70/89.50 |
| Teesta | 2014-04-22 | LC81380422014112LGN00 | L-8 | 25.70/89.50 |
| Teesta | 2014-10-31 | LC81380422014304LGN00 | L-8 | 25.70/89.50 |
| Teesta | 2014-12-02 | LC81380422014336LGN00 | L-8 | 25.70/89.50 |
| **Sentinel-1A** | | | | |
| Ganga | 2017-10-17 | S1A_IW_GRDH ⋯ _31A9_Cal_Spk_TC | S-1A | 24.83/87.92 |
| Ganga | 2018-07-20 | S1A_IW_GRDH ⋯ _BE68_Cal_Spk_TC | S-1A | 24.83/87.92 |
| Ganga | 2018-05-18 | S1A_IW_GRDH ⋯ _114F_Cal_Spk_TC | S-1A | 24.83/87.92 |
| Ganga | 2018-08-10 | S1A_IW_GRDH ⋯ _6C38_Cal_Spk_TC | S-1A | 24.83/87.92 |
| Ganga | 2018-09-06 | S1A_IW_GRDH ⋯ _4CBB_Cal_Spk_TC | S-1A | 24.83/87.92 |
| Ganga | 2016-04-13 | S1A_IW_GRDH ⋯ _4BF4_Cal_Spk_TC | S-1A | 24.83/87.92 |
| Ganga | 2018-01-18 | S1A_IW_GRDH ⋯ _831D_Cal_Spk_TC | S-1A | 24.83/87.92 |
| Ganga | 2017-09-27 | S1A_IW_GRDH ⋯ _2DC4_Cal_Spk_TC | S-1A | 24.08/89.03 |
| Ganga | 2018-05-13 | S1A_IW_GRDH ⋯ _EF71_Cal_Spk_TC | S-1A | 24.08/89.03 |
| Ganga | 2018-06-08 | S1A_IW_GRDH ⋯ _035D_Cal_Spk_TC | S-1A | 24.08/89.03 |
| Ganga | 2018-07-12 | S1A_IW_GRDH ⋯ _8CBA_Cal_Spk_TC | S-1A | 24.08/89.03 |
| Ganga | 2018-08-17 | S1A_IW_GRDH ⋯ _EF93_Cal_Spk_TC | S-1A | 24.08/89.03 |
| Brahamputra | 2018-07-14 | S1A_IW_GRDH ⋯ _2752_Cal_Spk_TC | S-1A | 25.18/89.66 |
| Brahamputra | 2017-11-14 | S1A_IW_GRDH ⋯ _BA85_TC_Cal | S-1A | 25.18/89.66 |
| Brahamputra | 2018-05-15 | S1A_IW_GRDH ⋯ _9533_Cal_Spk_TC | S-1A | 25.18/89.66 |
| Brahamputra | 2017-09-17 | S1A_IW_GRDH ⋯ _E022_Cal_Spk_TC | S-1A | 25.18/89.66 |
| Brahamputra | 2018-06-18 | S1A_IW_GRDH ⋯ _8D0F_Cal_Spk_TC | S-1A | 25.18/89.66 |
| Brahamputra | 2018-08-19 | S1A_IW_GRDH ⋯ _173D_Cal_Spk_TC | S-1A | 25.18/89.66 |
| Chenab | 2018-09-07 | S1A_IW_GRDH ⋯ _3240_Cal_Spk_TC | S-1A | 29.35/71.30 |
| Chenab | 2018-08-14 | S1A_IW_GRDH ⋯ _A3CB_Cal_Spk_TC | S-1A | 29.35/71.30 |
| Chenab | 2018-03-19 | S1A_IW_GRDH ⋯ _2E5E_Spk_TC | S-1A | 29.35/71.30 |
| Chenab | 2018-02-23 | S1A_IW_GRDH ⋯ _741E_TC_Cal_Spk | S-1A | 29.35/71.30 |

| | | | | |
|---|---|---|---|---|
| Indus | 2018-07-10 | S1A_IW_GRDH ⋯ _4B89_Cal_Spk_TC | S-1A | 25.35/68.35 |
| Indus | 2018-05-11 | S1A_IW_GRDH ⋯ _CE83_Cal_Spk_TC | S-1A | 25.35/68.35 |
| Indus | 2017-09-13 | S1A_IW_GRDH ⋯ _7DD5_Cal_Spk_TC | S-1A | 25.35/68.35 |
| Indus | 2018-08-27 | S1A_IW_GRDH ⋯ _DA34_Cal_Spk_TC | S-1A | 25.35/68.35 |
| Teesta | 2017-09-03 | S1A_IW_GRDH ⋯ _32D8_Cal_Spk_T | S-1A | 25.70/89.50 |
| Teesta | 2018-01-13 | S1A_IW_GRDH ⋯ _022569 | S-1A | 25.70/89.50 |
| Teesta | 2018-05-13 | S1A_IW_GRDH ⋯ _021886 | S-1A | 25.70/89.50 |
| Teesta | 2018-06-06 | S1A_IW_GRDH ⋯ _022236 | S-1A | 25.70/89.50 |
| Teesta | 2018-07-12 | S1A_IW_GRDH ⋯ _022761 | S-1A | 25.70/89.50 |
| Teesta | 2018-08-29 | S1A_IW_GRDH ⋯ _023461 | S-1A | 25.70/89.50 |
| Teesta | 2017-09-03 | S1A_IW_GRDH ⋯ _32D8_Cal_Spk_TC | S-1A | 26.33/88.87 |
| Teesta | 2018-01-13 | S1A_IW_GRDH ⋯ _50B1_Cal_Spk_TC | S-1A | 26.33/88.87 |
| Teesta | 2018-05-13 | S1A_IW_GRDH ⋯ _D8D7_Cal_Spk_TC | S-1A | 26.33/88.87 |
| Teesta | 2018-06-06 | S1A_IW_GRDH ⋯ _1753_Cal_Cal_TC | S-1A | 26.33/88.87 |
| Teesta | 2018-07-12 | S1A_IW_GRDH ⋯ _F499_Cal_Spk_TC | S-1A | 26.33/88.87 |
| Teesta | 2018-08-29 | S1A_IW_GRDH ⋯ _341E_Cal_Spk_TC | S-1A | 26.33/88.87 |
| Kosi | 2018-08-18 | S1A_IW_GRDH ⋯ _8CB2_Cal_Spk_TC | S-1A | 26.52/86.92 |
| Kosi | 2018-06-19 | S1A_IW_GRDH ⋯ _9B41_Cal_Spk_TC | S-1A | 26.52/86.92 |
| Kosi | 2017-04-25 | S1A_IW_GRDH ⋯ _32C5_Cal_Spk_TC | S-1A | 26.52/86.92 |
| Kosi | 2017-07-30 | S1A_IW_GRDH ⋯ _2658_Cal_Spk_TC | S-1A | 26.52/86.92 |

Table A1 Details of the Landsat-8 (L-8), Landsat-TM (L-TM) and Sentinel-1A (S-1A) satellite images used in this study. Gauge station is the location (lat/lon) of nearest in-situ discharge measurement stations.

## A2   Description of satellite and in-situ dataset

Table A2 contains a detailed description of in-situ discharge data for different rivers used in this study. This includes, data source, location of gauge station and period of measurement.

## A3   Glossary

## Appendix B:  Satellite image processing

### B0.1   Identification of river channels

To identify the river and non river pixels, we have used the infra-red bands of Landsat-TM and Landsat-8 images. In Landsat-TM, the infra-red $(0.76 - 0.90\,\mu\mathrm{m})$ wavelength corresponds to band 4 whereas, in Landsat-8 image, it corresponds to band 5 $(0.85 - 0.88\,\mu\mathrm{m})$.

| In-situ data (discharge) | | | | | | Satellite images (years) |
| River | Station | Location | | Period | Source | |
| | | Longitude (degree) | Latitude (degree) | | | |
| Teesta | Anderson | 88.87 | 26.33 | 1965-1971 | RivDIS | 2014, 2015, 2017, 2018 |
| | Kaunia | 89.50 | 25.70 | 1969-1975 | RivDIS | 2014, 2017, 2018 |
| Kosi | Bhimnagar | 86.92 | 26.52 | 2002-2014 | Kosi barrage Birpur | 1991, 1992, 2000, 2002 2011, 2013, 2015, 2017, 2018 |
| Chenab | Panjnad | 71.30 | 29.35 | 1973-1979 | RivDIS | 2013, 2014, 2015, 2018 |
| Indus | Kotri | 68.35 | 25.35 | 1973-1979 | RivDIS | 2014, 2015, 2017, 2018 |
| Ganga | Farakka | 87.92 | 24.83 | 1949-1973 1978-2007 | RivDIS & CWC | 2013, 2014, 2015, 2016, 2018 |
| | Paksay | 89.03 | 24.08 | 1965-1975 1996-2005 | RivDIS & Dhaka Univ. | 2009, 2013, 2014, 2017, 2018 |
| Brahmaputra | Bahadurabad | 89.66 | 25.18 | 1969-1975 1996-2005 | RivDIS & Dhaka Univ. | 2009, 2013, 2014, 2017, 2018 |

**Table A2.** Satellite images used for the extraction of channels width. In-situ discharge data is freely available and were downloaded from (http://www.rivdis.sr.unh.edu/maps/asi/).

We obtained an optimal threshold value by using the algorithm initially proposed by Yanni and Horne (1994). We than used the optimal threshold value to separate water and dry pixels from Landsat satellite images. The algorithm initiate by selecting a threshold as a midpoint value that lies in between the maximum and minimum gray level intensity (gi) as $gi_{mid} = (gi_{max} + gi_{min})/2$, where $gi_{max}$ is the highest and $gi_{min}$ is the lowest gray level intensity. Based on this initial threshold, the image pixels are clustered into foreground and background objects. After each iteration the threshold value is updated using the mean intensity of both the clusters. Finally the algorithm terminates when the threshold converges.

## B0.2   Removal of artefacts

Thresholding a gray scale input satellite image into binary class (water and dry pixels) produces spurious features. These consist of wet pixels that get classified as dry or of isolated water pixels that appear randomly in the binary images. Clusters (usually 2-3 pixels in size) that appear inside the river network do not corresponds to bars or islands. They appear to be more frequent in the areas where strong reflection from the bed sediment cause water pixels to appear more like a sand. Isolated water pixels that do not belong to the river are disconnected and located in water-logged areas. We have identified both of

| Type | Defination | Remarks | Source |
|---|---|---|---|
| Instantaneous discharge | discharge at any given time in space | usually measured at gauge stations installed on rivers | Chow (2010), Navratil et al. (2006) |
| Monthly average discharge | average discharge in a given month of the year | calculated by taking the mean of each month for the entire period of record | Chow (2010) |
| Annual average discharge | average discharge in a given year or time series | calculated by taking the mean of total discharge in a year or period | Chow (2010) |
| Median discharge | median value of discharge in a given year or period | calculated by finding the median value from the discharge time series of a given period | Blench (1957) |
| Bankfull discharge | discharge that completely fills the channel. | occurs once every year or in every two years | Wolman and Miller (1960), Navratil et al. (2006) (Rhoads, 2020, p. 145) |
| Formative discharge | derived discharge that would result in the same hydraulic geometry as the long-term hydrograph. | corresponds to average annual discharge, median flood discharge, and bankfull discharge | Leopold and Maddock (1953), Wolman and Miller (1960), Bolla Pittaluga et al. (2014) (Rhoads, 2020, p. 144) |

**Table A3.** A summary of terminology used for different discharges in this study.

these type errors from binary image and reprocess to remove them automatically. While doing this we first define a seed point inside the main channel and run the flood filling algorithm (Mehnert and Jackway, 1997; Bernander et al., 2013; Fan et al., 2005). This identify water pixels in a river channel those are connected and remove the isolated water pixels those have poor connectivity (Fig. B1).

**B0.3  Extraction of channel's skeleton and contour**

Our channel width extraction algorithm requires to river's centerline and boundary. A river centerline often called skeleton in computer vision corresponds to its median axis. To identify the river skeleton, we have used a thinning algorithm to extract river's centerline. The algorithm iteratively reduces the boundary pixels in a way that preserves its topology (for example eroding pixels must not alter the geometric properties of the object studied) and connectivity (Fig. B2 a). The final skeleton
is centered within the object and reflects its geometrical properties (Zhang and Suen, 1984; Baruch, 1988; Lam et al., 1992; Chatbri et al., 2015). The thinning algorithms produces several small centre line segments, often less than 300 meter in length, that are disconnected from the channel network at one end. These segments of the skeleton are too small to be considered as part of the river network. For our purpose we consider such segments as noise and filter them out. We do this iteratively, by looking for skeleton segments those are disconnected from the skeleton network at one end. To extract the channel banks, we

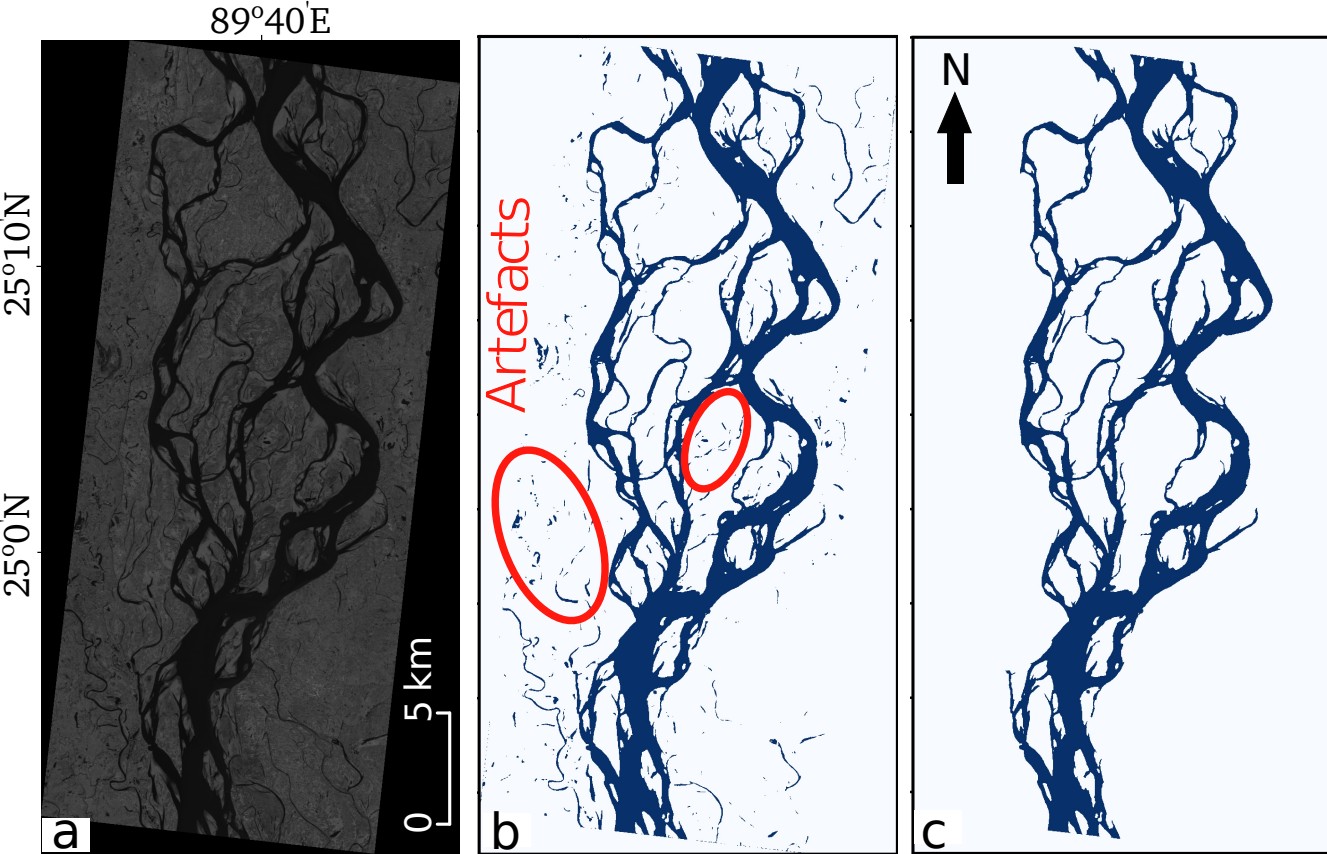

**Figure B1.** (a) Input Landsat-8 image of short wave infra-red wavelength is threshold to create binary image having water (blue) and dry (white) pixels. (b) Binary image with isolated water patches and artefacts and (c) removed artefacts (image source: Landsat-TM, 29 November 2013).

have applied a contour extraction algorithm that detects the outer boundary of a channel (Fig. B2). The algorithm relies on a pixel-neighbourhood analysis, where a pixel in a binary image is considered a contour pixel, if it has at least one background neighbour (Chatbri et al., 2015).

### B0.4    Channel's width calculation

Once the satellite images are processed to extract skeleton and channel's banks, we then proceed to extract the width of each
channels. We do this by measuring the distance from the centre of a channel to its banks orthogonally to the flow. From the skeleton of the image we draw a perpendicular line to the river bank and measure the the euclidean distance (Fig. B3). In case of a braided river, especially near the junctions where more than two river join or bifurcate form a complex network. At such locations our algorithm fails to measure correct width. To circumvent this we identify all the junctions from river skeleton

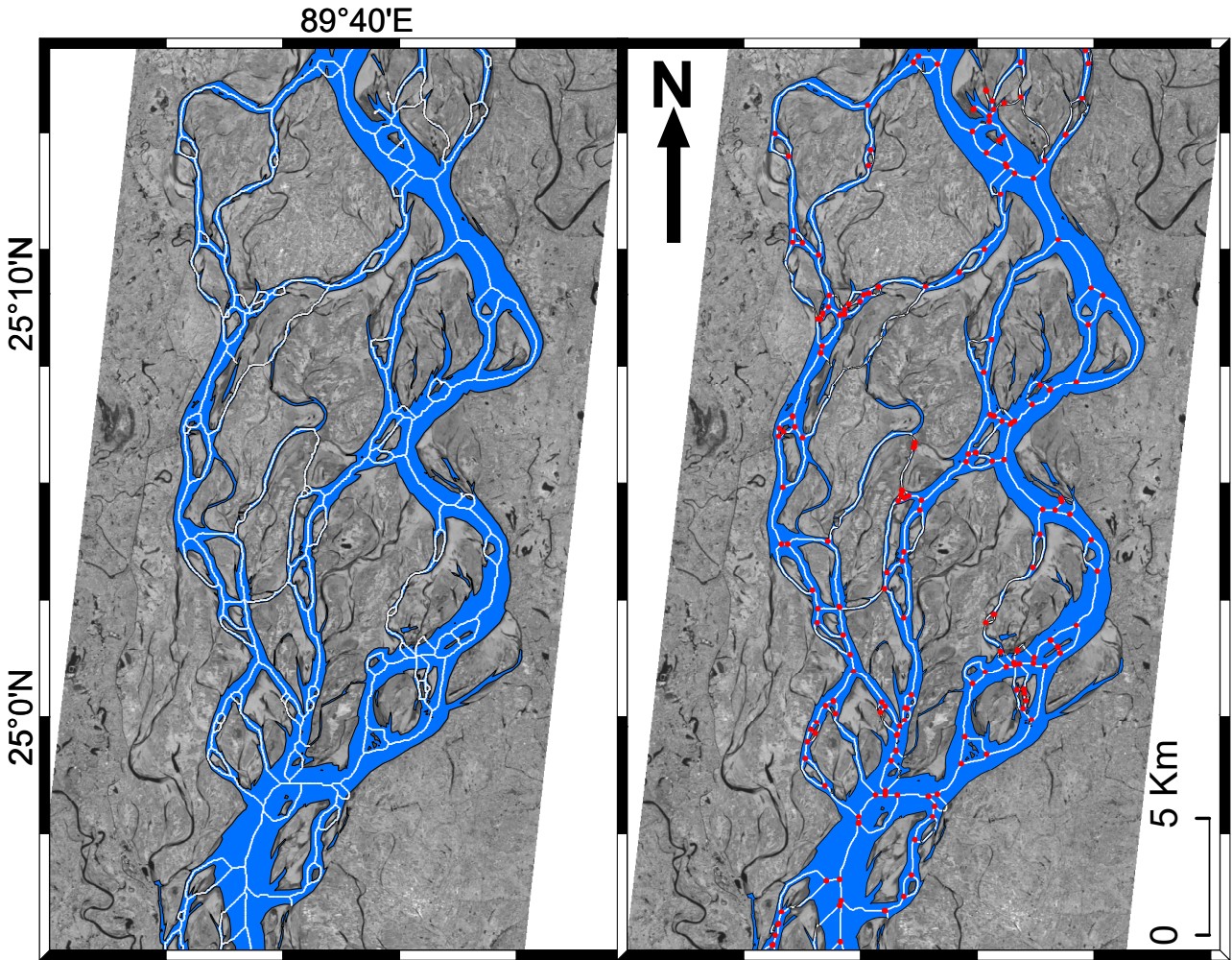

**Figure B2.** In left: river center line (skeleton) and boundary (contour) are superimposed on Landsat-8 satellite image. In right: image illustrates the stream junction identified on skeleton (image source: Landsat-TM, 29 November 2013).

(Fig. B2b). In the proximity of junction we consider an area of 5 pixels and define them as a zone of channel's confluence and
diffluence. In this zone we avoid to calculate the width of the channels.

Finally, we draw perpendicular transects from each pixel of the skeleton to both side of the channel and calculate the distance from any point $(x, y)$ on the skeleton to its corresponding left $(x_1, y_1)$ and right $(x_2, y_2)$ points on the channel boundary (Fig. B3). We then sum these widths to get the total width across a transect. For simplicity, at every one kilometer distance along the channel we compute the most probable width of each channels across a river section. Finally, the discharges through a section
can be calculated along an entire reach (Fig. B3).

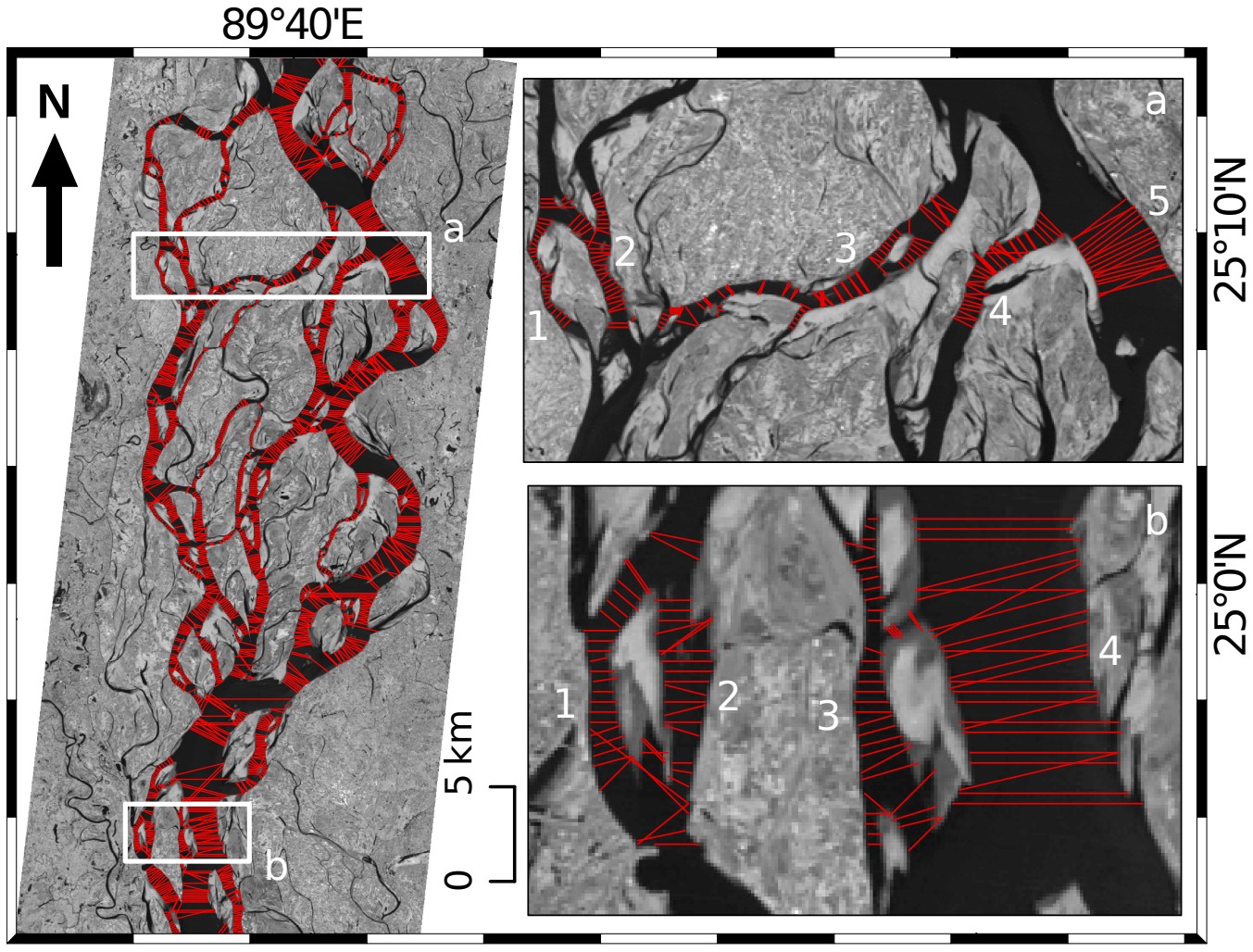

**Figure B3.** Width extracted across each of the individual channels. Image in the right illustrates the reach lengths (in boxes) over which most probable width of each channels is calculated (image source: Landsat-TM, 29 November 2013).

# Appendix C: Discharge estimation

## C1    Cross-section of a braided thread

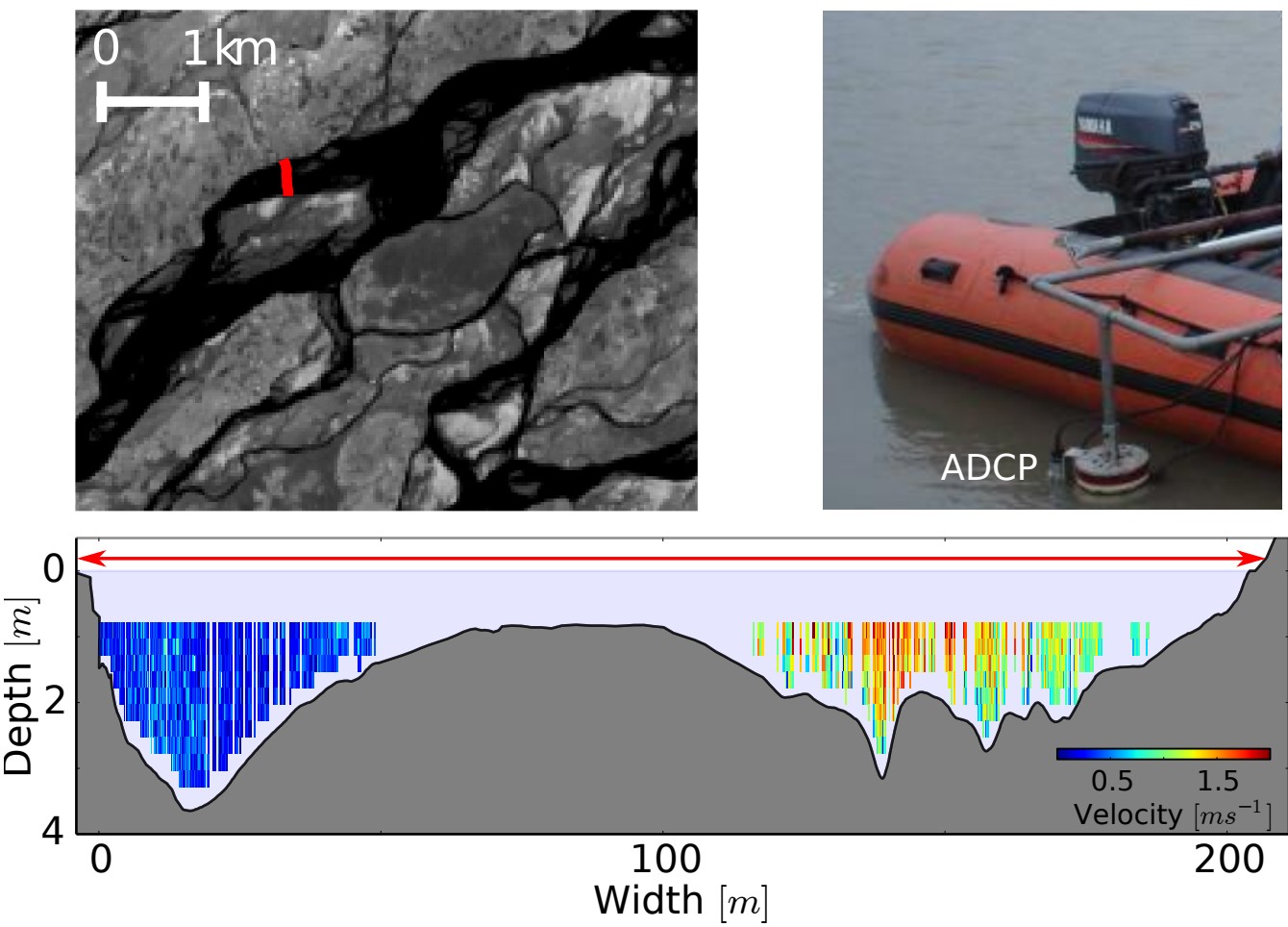

**Figure C1.** Velocity profile measured using and ADCP across a braided thread of the Kosi River in the Himalayan Foreland. Red horizontal line with arrow is to illustrate the top water surface. Color with different intensities show the magnitude of velocity ($\text{ms}^{-1}$)

## C2 Hydrograph of the Kosi River

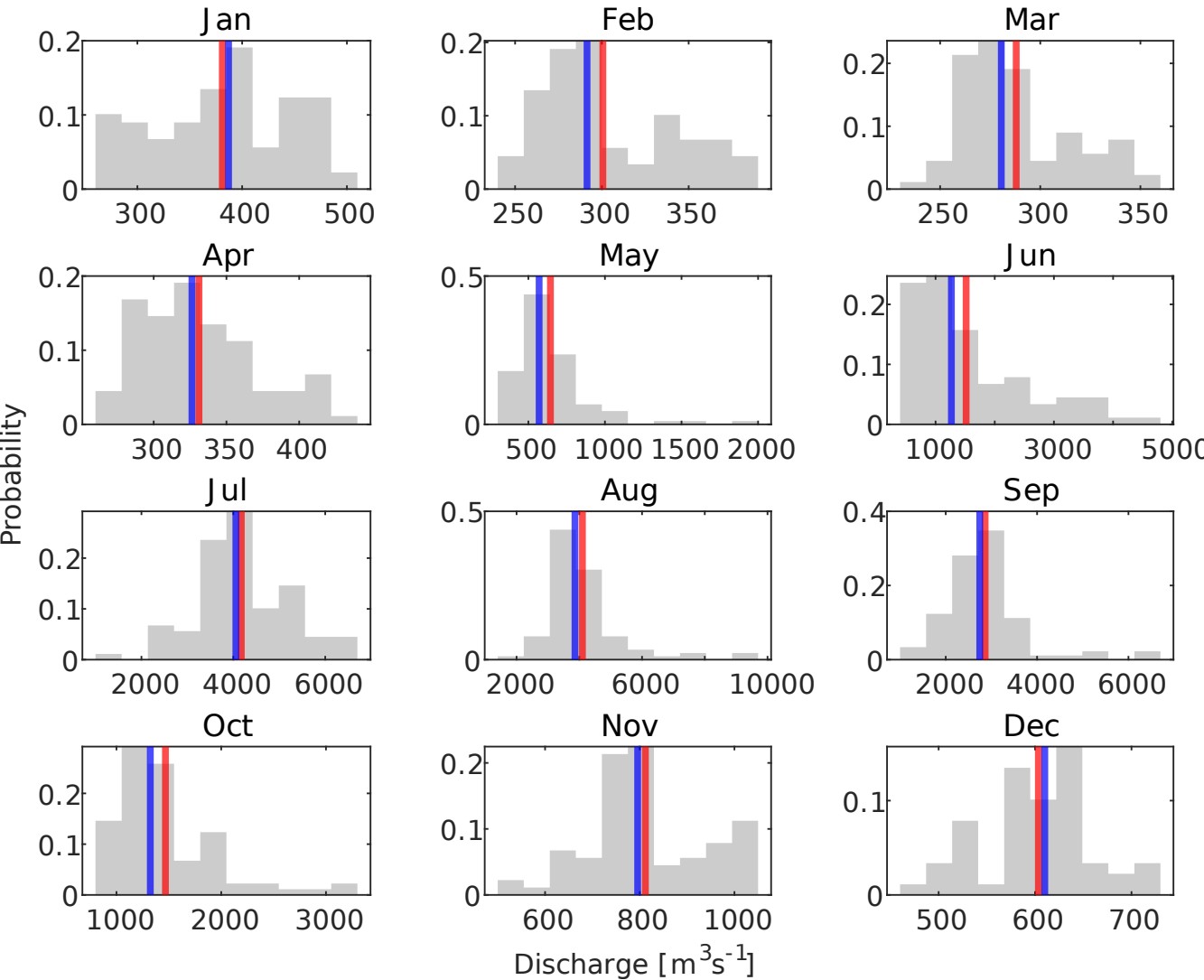

**Figure C2.** Histogram of daily discharge of the Kosi River measured at the Bhimnagar barrage in 2011, 2013, and 2014. Vertical lines in red and blue are the mean and median values of the probability distribution.

## C3 Evolution of discharge time series

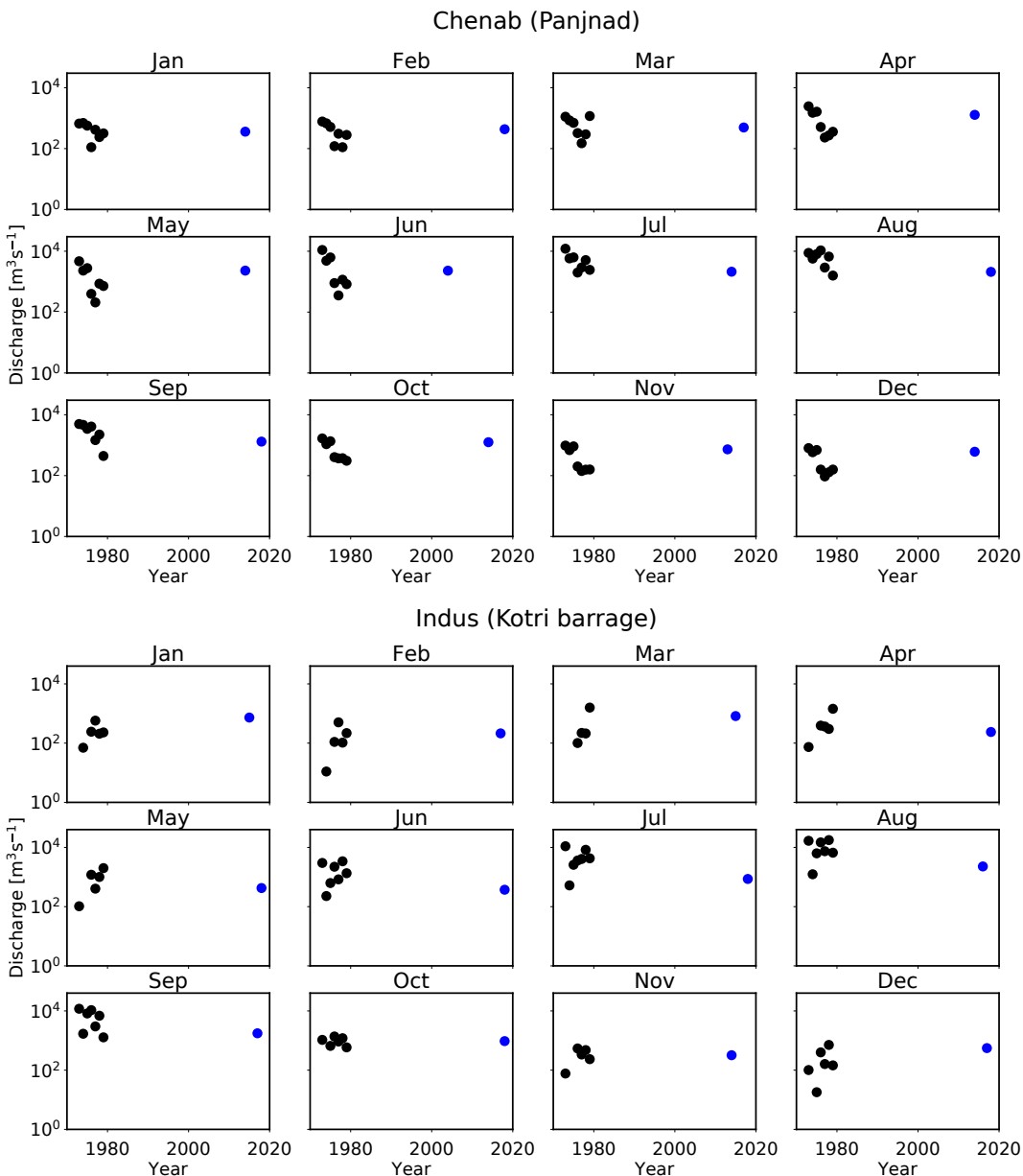

**Figure C3.** Time series of discharge of the Chenab and Indus river (circles in black) measured at the ground station (Panjnad and Kotri barrage). Circle in blue is the discharge estimated from satellite images

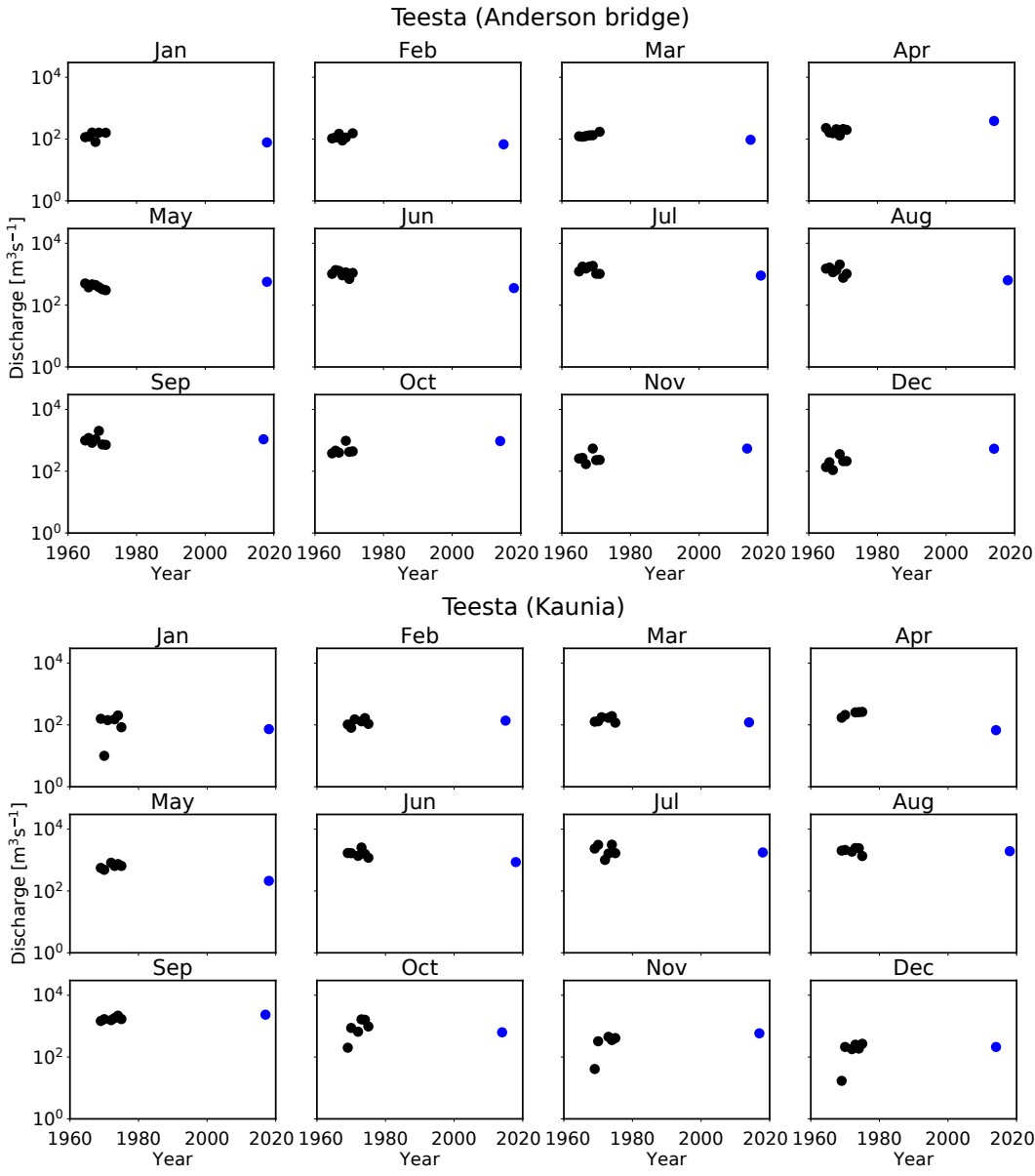

**Figure C4.** Time series of discharge of the Teesta River (circles in black) measured at the ground station (Anderson bridge and Kaunia). Circle in blue is the discharge estimated from satellite images.

## C4   Comparison of mean annual discharge with GRWL database

Allen and Pavelsky (2018) measured the width of the global rivers from Landsat images for the month when they commonly flow near mean discharge. In their database, Global River Width from Landsat (GRWL), for braided river they have reported the aggregated width of all the active threads. This width can not be used to estimate discharge from our regime curve that we established for the Himalayan Rivers. Our regime curve relates to the measurement of hydraulic geometry of individual threads of braided and meandering rivers (Gaurav et al., 2014, 2017), therefore it is applicable only at the thread scale. Since the resulting regime curve is non linear, estimating discharge across a transect in a braided river from the aggregated width will be different from the discharge obtained from the summation of discharge of the individual threads.

To overcome this, we have used binary water mask images from GRWL database to extract width of the individual threads. We then use these threads to estimate their discharge using our regime curve (equations. 4 and 6 in the manuscript). We observed for most of our rivers, discharge estimated from threads width extracted from the GRWL database falls within the same order of magnitude to the yearly average discharge measured at the corresponding gauge stations (Table C1).

| River | Station | $\langle Q_{insitu} \rangle$ $\mathrm{m^3 s^{-1}}$ | $\langle Q_{sat.} \rangle$ $\mathrm{m^3 s^{-1}}$ | $\langle Q_{GRWL} \rangle$ $\mathrm{m^3 s^{-1}}$ |
|---|---|---|---|---|
| Teesta | Anderson | $605 \pm 109$ | $638 \pm 165$ | $408 \pm 177$ |
| Teesta | Kaunia | $924 \pm 144$ | $745 \pm 155$ | $400 \pm 110$ |
| Kosi | Bhimnagar | $1559 \pm 313$ | $1810 \pm 380$ | $2936 \pm 625$ |
| Chenab | Panjnad | $2500 \pm 961$ | $1275 \pm 268$ | $937 \pm 344$ |
| Indus | Kotri | $3745 \pm 825$ | $794 \pm 162$ | $218 \pm 102$ |
| Ganga | Farakka | $11477 \pm 2279$ | $10593 \pm 2225$ | $15959 \pm 9616$ |
| Ganga | Paksay | $12080 \pm 2403$ | $11605 \pm 2438$ | $5679 \pm 3310$ |
| Brahmaputra | Bahadurabad | $21751 \pm 2942$ | $21717 \pm 4740$ | $11149 \pm 5122$ |

**Table C1.** Annual average discharge measured at the gauge station and estimated from satellite images. $\langle Q_{GRWL} \rangle$ is the discharge estimated from binary water mask from GRWL database from Allen and Pavelsky (2018).

*Author contributions.* KG, AVS and FM have conceptualised the study, KG collected the field data, AVS and KG developed the algorithm to process satellite images, AK has processed the Sentinel-1 satellite images. KG has written the manuscript and FM, RS and SKT have reviewed. All authors discussed the results and contributed to the final manuscript.

*Competing interests.* The authors declare that they have no conflict of interest

*Acknowledgements.* K.G and AV.S acknowledge the Ministry of Earth Sciences, Government of India for research funding through Letter no.: MoES/PAMC/H & M/84/2016-PC-II. K.G would like thank Dr. Olivier Devauchelle for fruitful discussion. We thank Mr. Hasnat Jaman, a former student of the Geology department of University of Dhaka for providing the discharge data for the Ganga and Brahmaputra rivers. Satellite imagery courtesy of USGS/NASA and ESA. We are thankful to Central Water commission, Ministry of water resources, New Delhi, Engineers of the Investigation and research division of the Kosi river Project and global river discharge databases (RivDIS v1.0) for providing the in-situ measurement of discharge

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
