# Peer review of "Coupling threshold theory and satellite image derived channel width to estimate the formative discharge of Himalayan Foreland rivers."

_Earth Surface Dynamics, 2020_

## Referee Comment (RC1) · Kieran Dunne (Referee) · 7 Sep 2020

General Comments:

This manuscript presents a novel, potentially quite powerful methodology to extract the formative discharges of ungauged, alluvial rivers, utilizing a combination of an innovative remote sensing technique coupled with a mechanistically-based relationship between river channel width and water discharge from threshold channel theory. Overall, I like this paper. I think it demonstrates a strong linkage between analysis of remotely sensed data and mechanistic theory, allowing for improved understanding of the processes at play in environment where it might be more difficult to employ the standard

suite of direct empirical measurements – fluvial geomorphologists working on Martian channels has been doing this for years quite successfully. I find this manuscript to be in good shape overall. I have do have a few minor questions/clarifications that I have outlined below:

Specific comments:

Line 98: When you are defining your variables, you set your threshold Shields parameter equal to 0.3. This is an order of magnitude greater that the more standard range (0.03-0.05) that is usually observed for grains/channels under the flow conditions found in your typical natural river. I am not criticizing the usage of this value in the model, but I believe that it would be beneficial to readers to clarify this discrepancy. The first two authors have published papers where they employed the same threshold model on rivers on the Kosi Megafan and rivers in the Bayanbulak Grassland (this reviewer is incredibly envious of their field sites!), but have used threshold Shields parameter values of 0.3 and 0.04, respectively. Given that the critical shear stress/shields stress is a physical parameter that can be either measured directly or calculated based upon measurable gain size data, I believe that it would be worth explaining departure from the more commonly used 0.03-0.05 values, or at least stating that the usage of this offset 0.3 value has been shown to be effective for explaining the geometry of the category of rivers that the ones used in this study fall under.

Figure 4: The panes that should show both the binary and raw images are empty, am I missing something? Maybe just my computer acting up, but I tried downloading the PDF a few times with no effect.

Line 200: How are the histograms skewed? Is the skewness a result of natural variation in thread width or error in the cross-section selection?

Line 201: What is meant by "post probable?" Median? Modal?

Line 210: Could you explain a bit more how you got from equation 4 to equation 6,

even if you put it in the appendices? Where does the sqrt(g d) come from?

Line 230: Okay so here is where I start to reflect and have a few structural problems with the paper. I think a lot of this material discussing the formative discharge and its control on channel morphology needs to be made earlier on in the paper, either in the introduction or at the point where the authors introduce equation 4. I found myself a bit confused when I was reading the results section (specifically Fig. 8) where estimates of monthly discharge were being made within a threshold channel geometry theoretical framework that isn't really meant to reflect the month to month flow width-discharge relationship, and it took me a while to realize that the main point is that the model does a good job at recognizing formative discharges, but does not do so well when it comes to recognizing discharges below that. I think that the clarity of the manuscript could be improved if the authors clearly introduced earlier on that the goal of the remote sensing analysis coupling with theory would be to identify the formative discharge of the channels. I think this might clarify to the reader exactly what their coupling of threshold theory and satellite imagery analysis is capable of producing.

---

## Referee Comment (RC2) · Anonymous Referee #2 · 29 Sep 2020

Summary. The authors utilize hydrograph records and satellite imagery to develop algorithms where discharge can be predicted based on the formative width of the river channel.

General comments. Overall, I have few comments as the paper is well written and conveys its results in a straightforward manner. I believe with very minor revisions this manuscript would be suitable for publication within Earth Surface Dynamics. The details provided on additional data and the methodology within the appendix are welcome.

There are a few points throughout that may not be well established. Namely, that the

extracted width during the wet season is a good proxy for the formative discharge. The idea that the formative discharge occurs at annual timescales has always felt like a misinterpretation of Leopold and Maddock and Wolman and Miller. It occurs at annual to multiple year timescales in temperate regions based on the frequency of events. Arid regions flood considerably less. The reasonable match between discharge from Equation 4 & 6 suggests that the flow is likely close to the formative value, but some discussion on the recurrence of formative flows within this system would be welcome.

Another point of concern, though fairly noted by the authors, that could use more discussion is the non overlapping satellite images and discharge records. For some rivers they are fairly close, but others like the Chenab and Teesta have discharge records from the 1970s and images from 2014 and 2018, respectively. It is not clear that averages taken from the 1970s should be compared with measurements from current times without significant effort to establish that the underlying timeseries is non-stationary. The acknowledgement of the changes to flow due to anthropogenic modification is a step in this direction, however the step from width to discharge relies on the idea that the river is self-forming, which may not be the case in many managed and modified large rivers. If the timescale of river adjustment is relatively quick then the formative discharge always matches the width, however if adjustment to modification of the flow or climate change is slow the formative width concept estimates will lag the actual discharge. I would greatly appreciate the authors providing more insight into these issues within the discussion.

A note on timeseries here. Is the monthly mean value representative of the actual hydrograph? The rivers are relatively large and that may be the case, however I would feel more at ease with the methodology if I knew that the hydrographs were not being under sampled to a degree that they may not adequately represent the flow in the system anymore.

To better understand the utility of this method relative to other existing methods the authors should consider a comparison with the data available from Allen and Pavelsky

**ESurfD**
which presumably covers many of the same rivers. Their method and the authors have similar data limitations and therefore would inspire more confidence in the current assessment and indicate potential broader applicability of the methods developed here.

Specific Comments.

Ln. 65 - A few lines explaining what the 'threshold theory' entails would be welcome here. What does the threshold theory say about rivers that allows this method to progress?

Table A2 - Could you add a column listing the years of satellite images on the right. These data are in table A1 as a list, but a summary would be welcome here.

---

## Editor Comment (EC1) · Rebecca Hodge (Editor) · 30 Oct 2020

Sorry for the lateness of this comment. The two reviewers have provided several useful points for you to think about, and I encourage you to address them fully. I wanted to add a further comment for you to consider when revising your paper. The discussion currently focusses on the question of whether the identified discharge is a formative discharge. However, it would be beneficial for it also to address some wider issues, which will help those that might be interested in applying these methods elsewhere. For example, are there any errors or uncertainties associated with the methods that it is important to consider? What sort of datasets are required? Secondly, could these

methods be applied elsewhere? Under what conditions would this method work or not work?

---

## Author Response (AR1)

November 03, 2020

Dear Editor,

On the behalf of co-authors, I would like to thank the reviewers and the associated editor for their constructive comments. Their suggestions and comments were very useful to improve the overall quality of the manuscript. Following the reviewer's comments, we have answered them point-by-point and revised the manuscript accordingly.

We have enclosed the following with this letter ;

— a detailed response to both the reviewers

— a difference file showing the changes in the initial version of the manuscript in response to the comments

I hope this revised manuscript will be considered suitable for publication in Esurf.

Your truly

Kumar Gaurav

**Answer to Dr.Kieran Dunne**

Text in black is the comments from referees

Text in blue is the author's response

**General comments:**

This manuscript presents a novel, potentially quite powerful methodology to extract the formative discharges of ungauged, alluvial rivers, utilizing a combination of an innovative remote sensing technique coupled with a mechanistically-based relationship be-tween river channel width and water discharge from threshold channel theory. Overall,I like this paper. I think it demonstrates a strong linkage between analysis of remotely sensed data and mechanistic theory, allowing for improved understanding of the processes at play in environment where it might be more difficult to employ the standard suite of direct empirical measurements - fluvial geomorphologists working on Martian channels has been doing this for years quite successfully. I find this manuscript to be in good shape overall. I have do have a few minor questions/clarifications that I have outlined below

**Specific comments:**

Line 98: When you are defining your variables, you set your threshold Shields parameter equal to 0.3. This is an order of magnitude greater that the more standard range(0.03-0.05) that is usually observed for grains/channels under the flow conditions found in your typical natural river. I am not criticizing the usage of this value in the model, but I believe that it would be beneficial to readers to clarify this discrepancy. The first two authors have published papers where they employed the same threshold model on rivers on the Kosi Megafan and rivers in the Bayanbulak Grassland (this reviewer is incredibly envious of their field sites!), but have used threshold Shields parameter values of 0.3and 0.04, respectively. Given that the critical shear stress/shields stress is a physical parameter that can be either measured directly or calculated based upon measurable grain size data, I believe that it would be worth explaining departure from the more commonly used 0.03-0.05 values, or at least stating that the usage of this offset 0.3 value has been shown to be effective for explaining the geometry of the category of rivers that the ones used in this study fall under.

We agree that the value we use is large and acknowledge that there is a misunderstanding here. We have clarified this in section 3 (lines 100-106) in the revised manuscript. The sediment is in the fine sand-silt range and thus one expects values of the critical shields stress to be much higher then the 0.03-0.04, value commonly used for gravel bed rivers. We propose to add the following paragraph to explain our choice of value.

"Typical grain size of the sediments of the Himalayan Foreland rivers is order of $d_s = 100 - 300\mu m$. Thus the dimensionless grain size reads;

$$D^* = \left(\frac{d_s^3 g \rho_s^2}{\eta^2}\right)^{1/3} \simeq 1 - 6$$

where $g = 9.81\ m/s^2$ is the acceleration of gravity, $\rho_s = 2650\ kg/m^3$ is the sediment density, and $\eta = 10^{-3}\ Pa.s$ is the dynamic viscosity of water. In this range of values the critical shields number $\theta_t$ is on order of $\theta_t \sim 0.1$ with a maximum around 0.3 (Julien, 1995; Selim Yalin, 1992). Delorme et al. (2017) recently obtained an experimental value of $\theta_t \sim 0.25$ for silica sands of size 150 $\mu m$. We therefore took the upper value of 0.3 as a conservative estimate. Taking lower values of $\theta_t$ such as the classical 0.1 would lead to a slightly better match between the theoretical prediction and the data but does not lead to any significant change in our conclusions."

Figure 4: The panes that should show both the binary and raw images are empty, am I missing something? Maybe just my computer acting up, but I tried downloading the PDF a few times with no effect.

This is probably an issue of image format. In the revised manuscript, we have replaced this image with an appropriate format.

Line 200: How are the histograms skewed? Is the skewness a result of natural variation in thread width or error in the cross-section selection?

To address this we have modified section (5.2, lines 218-220) in the revised manuscript to explain skewness of the width histogram. This skewness mainly results from the natural variability of width along the threads and also due to the error in the cross-section extracted from images, particularly at the location where curvature of a threads is high. We have also included the histogram shown below (Figure 1) in the revised manuscript (Fig.8) to show the distribution of threads width in a braided river.

Line 201: What is meant by "post probable?" Median? Modal?

Since the width of threads in braided rivers varies significantly along its course, their probability distributions are skewed. To take this skewness into account, we use the most probable width ($W_m$) as a representative value of the width. This value corresponds to a geometric mean of the distribution. To illustrate this we have included the figure shown below (Figure1) in the revised manuscript (Fig.8) to illustrate the probability distribution of width in a braided river. The red vertical line shows the geometrical mean of thread's width.

Line 210: Could you explain a bit more how you got from equation 4 to equation 6,even if you

[Figure]

Figure 1: Distribution of thread's width measured across two different reaches in a braided river. Vertical line in red shows the most representative width that corresponds to geometric mean.

put it in the appendices? Where does the sqrt(gd) come from?

Thank you for highlighting this. We have rewritten the equation 4 in the dimensionless form as given below. This clarifies how equation 6 is obtained from equation 4.

$$\frac{W}{d_s} = \left[ \frac{\pi}{\mu} \left( \frac{\theta_t(\rho_s - \rho_f)}{\rho_f} \right)^{0.25} \sqrt{\frac{3C_f}{2^{3/2}\mathcal{K}\left[1/2\right]}} \right] Q_*^{0.5} \tag{1}$$

where $Q_* = Q_w/(d_s^2\sqrt{gd_s})$ is the dimensionless water discharge, $d_s$ is the grain size, $\rho_f \approx 1000\,\mathrm{kg\,m^{-3}}$ is the density of water, $\rho_s \approx 2650\,\mathrm{kg\,m^{-3}}$ is the density of quartz, $g \approx 9.81\,\mathrm{m\,s^{-2}}$ is the acceleration of gravity, $C_f \approx 0.1$ is the Chézy friction factor, $\mu \approx 0.7$ is the Coulomb's coefficient of friction, $\mathcal{K}(1/2) \approx 1.85$ is the elliptic integral of the first kind, and $\theta_t \approx 0.3$ is the

threshold Shield's parameter.

Line 230: Okay so here is where I start to reflect and have a few structural problems with the paper. I think a lot of this material discussing the formative discharge and its control on channel morphology needs to be made earlier on in the paper, either in the introduction or at the point where the authors introduce equation 4. I found myself a bit confused when I was reading the results section (specifically Fig. 8) where estimates of monthly discharge were being made within a threshold channel geometry theoretical framework that isn't really meant to reflect the month to month flow width-discharge relationship, and it took me a while to realize that the main point is that the model does a good job at recognizing formative discharges, but does not do so well when it comes to recognizing discharges below that. I think that the clarity of the manuscript could be improved if the authors clearly introduced earlier on that the goal of the remote sensing analysis coupling with theory would be to identify the formative discharge of the channels. I think this might clarify to the reader exactly what their coupling of threshold theory and satellite imagery analysis is capable of producing.

Thank you for the suggestion. We have added a paragraph in section 3 (lines 107-115) to bring more clarity on the research problem we are addressing. In this section, we have briefly introduced the concept of formative discharge that forms the geometry of natural alluvial rivers. Further we have highlighted how our knowledge of the morphology of a threshold channel can lead us to assess discharge that sets the geometry of natural alluvial channels by using thread's width derived from remote sensing images.

**Answer to the reviewer: 2**

Summary. The authors utilize hydrograph records and satellite imagery to develop algorithms where discharge can be predicted based on the formative width of the river channel.

**General comments:** Overall, I have few comments as the paper is well written and conveys its results in a straightforward manner. I believe with very minor revisions this manuscript would be suitable for publication within Earth Surface Dynamics. The details provided on additional data and the methodology within the appendix are welcome.

We are thankful to the reviewer for finding this manuscript suitable for publication in Earth Surface Dynamics.

There are a few points throughout that may not be well established. Namely, that the extracted width during the wet season is a good proxy for the formative discharge. The idea that the formative discharge occurs at annual timescales has always felt like a misinterpretation of Leopold and Maddock and Wolman and Miller. It occurs at annual to multiple year timescales in temperate regions based on the frequency of events. Arid regions flood considerably less. The reasonable match between discharge from Equation 4 & 6 suggests that the flow is likely close to the formative value, but some discussion on the recurrence of formative flows within this system would be welcome.

In order to clarify our analysis we added the following paragraph in the discussion section (lines 263-268).

"Formative discharge in the Himalayan Rivers occurs during the monsoon period at annual time scale (Roy and Sinha, 2014). This clearly reflects in the discharge hydrographs estimated from the measurement of channel's width from the satellite images. Furthermore, Métivier et al. (2017) have recently shown that non cohesive streams laden with sediments cannot have a width much larger than the width of a threshold stream before they start to braid. They also showed that, for experimental braided rivers, threads are always formed at bankfull flow, and at the limit of stability. Our hypothesis is thus that the formative discharge of threads in the Ganga plain is the bankfull discharge."

Another point of concern, though fairly noted by the authors, that could use more discussion is the non overlapping satellite images and discharge records. For some rivers they are fairly close, but others like the Chenab and Teesta have discharge records from the 1970s and images from 2014 and 2018, respectively. It is not clear that averages taken from the 1970s should be compared with measurements from current times with-out significant effort to establish that the underlying time series is non-stationary. The acknowledgement of the changes to flow

due to anthropogenic modification is a step in this direction, however the step from width to discharge relies on the idea that the river is self-forming, which may not be the case in many managed and modified large rivers. If the timescale of river adjustment is relatively quick then the formative discharge always matches the width, however if adjustment to modification of the flow or climate change is slow the formative width concept estimates will lag the actual discharge. I would greatly appreciate the authors providing more insight into these issues within the discussion.

We agree with the reviewer. To address this, we plot the time series of monthly discharge recorded at the gauge station (1973-1979) and estimated from satellite images (2014-2018) for the Indus, Chenab, and Teesta rivers (Figures 2 & 3, 4 and 5). Despite a large variability, the discharge time series of Indus and Chenab rivers show a strong declining trend during the monsoon period (June-September). Whereas discharge during the non-monsoon period appears to remain constant around the mean. Figures (2 & 3) clearly show that discharge estimated from satellite images plot within the variability of the observed trend. The estimated discharge of the Teesta River also plot within the noise of observed trends (Figures 4 and 5).

Further, alluvial rivers adjust their width at relatively short time scale in response to the formative discharge. Métivier et al. (2017) suggested a "limit-channel width", a largest possible width of a stable alluvial channel. Beyond this value, channel destabilizes into a braid and readjusts the width until they reach the "limit-width". On an average this process occurs at the annual time scale in the Himalayan Foreland rivers. In the revised manuscript we have added a paragraph (lines 304-310) in discussion section to explain the monthly discharge time series of the Indus, Chenab and Teesta rivers. We have also included the figures (C2 and C2) of discharge time series in the Appendix C.

A note on time series here. Is the monthly mean value representative of the actual hydrograph? The rivers are relatively large and that may be the case, however I would feel more at ease with the methodology if I knew that the hydrographs were not being under sampled to a degree that they may not adequately represent the flow in the system anymore.

For most of our rivers we could only access the monthly discharge recorded at the ground station. This is why we have considered the average monthly discharge to compare the discharge estimated from satellite images. To test whether this value is a representative of the actual hydrograph, we explore the daily in-situ discharge data of the Kosi River for years 2011, 2013, and 2014. Figure 6 shows the probability distribution of the discharge values recorded in different months. The vertical lines in red and blue are the mean and median values. Figure 6 clearly shows that the mean and median of the distribution of discharge are closely placed. This

[Figure]

Figure 2: Time series of discharge of the Indus River (circle in black) measured at the ground station (Kotri). Circle in blue is the discharge estimated from satellite images.

[Figure]

Figure 3: Time series of discharge of the Chenab River (circle in black) measured at the ground station (Panjnad). Circle in blue is the discharge estimated from satellite images.

[Figure]

Figure 4: Time series of discharge of the Teesta River (circle in black) measured at the ground station (Anderson bridge). Circle in blue is the discharge estimated from satellite images.

[Figure]

Figure 5: Time series of discharge of the Teesta River (circle in black) measured at the ground station (Kaunia). Circle in blue is the discharge estimated from satellite images.

suggests monthly average discharge of the Himalayan Foreland can be used as a representative value of actual hydrograph. In the revised manuscript we have included this hydrograph in in Appendix C (Figure C1) and also added a line (line 282) in the discussion section that the monthly average discharge appears to be a representative value of the actual hydrograph in the Himalayan Foreland.

[Figure]

Figure 6: Histogram of daily discharge of the Kosi River measured at the Bhimnagar barrage in 2011, 2013, and 2014. Vertical lines in red and blue are the mean and median values of the probability distribution.

To better understand the utility of this method relative to other existing methods the authors should consider a comparison with the data available from Allen and Pavelsky which presumably covers many of the same rivers. Their method and the authors have similar data limitations and therefore would inspire more confidence in the current assessment and indicate potential broader applicability of the methods developed here.

In order to address this, in the revised manuscript we have included the following analysis in the discussion section (lines 311-320). Allen and Pavelsky (2018) measured the width of the global rivers from Landsat images for the month when they commonly flow near mean discharge. In their database, Global River Width from Landsat (GRWL), for braided river they have reported the aggregated width of all the active threads. This width can not be used to estimate discharge from our regime curve that we established for the Himalayan Rivers. Our regime curve relates to the measurement of hydraulic geometry of individual threads of braided and meandering rivers (Gaurav et al., 2015, 2017), therefore it is applicable only at the thread scale. Since the resulting regime curve is non linear, estimating discharge across a transect in a braided river from the aggregated width will be different from a discharge obtained from the summation of discharge of the individual threads.

To overcome this we have used binary water mask images from GRWL database to extract width of the individual threads. We then use these threads to estimate their discharge using our regime curve (equations. 4 and 6 in the manuscript). We observed for most of our rivers, discharge estimated from threads width extracted from the GRWL database falls within the same order of magnitude to the yearly average discharge measured at the corresponding gauge stations (Table 1). This suggests that water mask from GRWL database can be used as a first order approximation of the mean discharge of the Himalayan Foreland rivers. We noticed that the discharge estimated from GRWL database appears more likely to occur during the early (Jun, July) or late (Sept, Oct) monsoon. Also we have included the (Table 1) in appendix C (Table C1) in the revised manuscript.

**Specific comments:** Ln. 65- A few lines explaining what the 'threshold theory' entails would be welcome here. What does the threshold theory say about rivers that allows this method to progress?

We have added a paragraph (lines 109-116) on threshold theory and its applicability in explaining the morphology of a sandy bed alluvial rivers.

Table A2-Could you add a column listing the years of satellite images on the right.These data are in table A1 as a list, but a summary would be welcome here.

Done.

| River | Station | $\langle Q_{\text{insitu}} \rangle$ | $\langle Q_{\text{sat.}} \rangle$ | $\langle Q_{\text{GRWL}} \rangle$ |
| --- | --- | --- | --- | --- |
| | | $\text{m}^3\text{s}^{-1}$ | $\text{m}^3\text{s}^{-1}$ | $\text{m}^3\text{s}^{-1}$ |
| Teesta | Anderson | $605 \pm 109$ | $638 \pm 165$ | $408 \pm 177$ |
| Teesta | Kaunia | $924 \pm 144$ | $745 \pm 155$ | $400 \pm 110$ |
| Kosi | Bhimnagar | $1559 \pm 313$ | $1810 \pm 380$ | $2936 \pm 625$ |
| Chenab | Panjnad | $2500 \pm 961$ | $1275 \pm 268$ | $937 \pm 344$ |
| Indus | Kotri | $3745 \pm 825$ | $794 \pm 162$ | $218 \pm 102$ |
| Ganga | Farakka | $11477 \pm 2279$ | $10593 \pm 2225$ | $15959 \pm 9616$ |
| Ganga | Paksay | $12080 \pm 2403$ | $11605 \pm 2438$ | $5679 \pm 3310$ |
| Brahmaputra | Bahadurabad | $21751 \pm 2942$ | $21717 \pm 4740$ | $11149 \pm 5122$ |

Table 1: Annual average discharge measured at the gauge station and estimated from satellite images. $\langle Q_{GRWL} \rangle$ is the discharge estimated from binary water mask from GRWL database from Allen and Pavelsky (2018)

**Answer to Dr. Rebecca Hodge**

Sorry for the lateness of this comment. The two reviewers have provided several useful points for you to think about, and I encourage you to address them fully. I wanted to add a further comment for you to consider when revising your paper. The discussion currently focusses on the question of whether the identified discharge is a formative discharge. However, it would be beneficial for it also to address some wider issues, which will help those that might be interested in applying these methods elsewhere. For example, are there any errors or uncertainties associated with the methods that it is important to consider? What sort of datasets are required? Secondly, could these methods be applied elsewhere? Under what conditions would this method work or notwork?

In order to address this we have added the following paragraph in the conclusion.

"One of the main source of uncertainty in discharge estimate is due to the error in the measurement of thread's width. This depends on the image resolution and the accuracy of the algorithm used to classify the river pixels from remote sensing images. A better resolution remote sensing images would most likely minimise the uncertainty and improve the agreement between estimated and in-situ discharge. Further our regime equation established for Himalayan rivers is based on a simple physical mechanism that explains the geometry of alluvial channels. We therefore suspect that the procedure we have established could be extended to most alluvial rivers. Globally it has been observed that the threshold theory well predicts the exponent of the regime equation (Eq. 4, in the manuscript), however the prefactor may vary significantly depending on the grain size distribution, turbulent friction coefficient and the critical shield parameter (Métivier et al., 2017). It is therefore suggested to modify this regime curve from the measurement of width, discharge and grain size of a individual thread's of braided and meandering channels in the field before applying it to the rivers of different climatic regime. Further it should be noted that our regime curve relates to the measurement of hydraulic geometry of individual threads of braided and meandering rivers, therefore it is applicable only at the thread scale. Since the resulting regime curve is non linear, estimating discharge across a transect in a braided river from the aggregated width will be different from the one obtained after the summation of discharges of the individual threads."

**Coupling threshold theory and satellite image derived channel width to estimate the formative discharge of Himalayan Foreland rivers.**

Kumar Gaurav[1], François Métivier[2], AV Sreejith[3], Rajiv Sinha[4], Amit Kumar[1], and Sampat Kumar Tandon[1]

[1]Indian Institute of Science Education and Research, Bhopal,462066, M.P, India
[2]Institute de Physique du Globe de Paris, 1 Rue Jussieu, 75005 Paris cedex 05, France
[3]School of Mathematics and Computer Science, Indian Institute of Technology, Goa, 403401, Goa, India
[4]Department of Earth Sciences, Indian Institute of Technology, Kanpur, 208016 UP, India

**Correspondence:** K.Gaurav (kgaurav@iiserb.ac.in)

**Abstract.** We propose an innovative methodology to estimate the formative discharge of alluvial rivers from remote sensing images. This procedure involves automatic extraction of the width of a channel from Landsat Thematic Mapper, Landsat 8, and Sentinel-1 satellite images. We translate the channel width extracted from satellite images to discharge by using a width-discharge regime curve established previously by us for the Himalayan Rivers. This regime curve is based on the threshold theory, a simple physical force balance that explains the first-order geometry of alluvial channels. Using this procedure, we estimate the discharge of six major rivers of the Himalayan Foreland: the Brahmaputra, Chenab, Ganga, Indus, Kosi, and Teesta rivers. Except highly regulated rivers (Indus and Chenab), our estimates of the discharge from satellite images can be compared with the mean annual discharge obtained from historical records of gauging stations. We have shown that this procedure applies both to braided and single-thread rivers over a large territory. Further our methodology to estimate discharge from remote sensing images does not rely on continuous ground calibration.

**Keywords:** Himalayan Foreland; regime curve; threshold theory; formative discharge

*Copyright statement.* This is an open access article under the terms of the Creative Commons Attribution License, which permits use, distribution and reproduction in any medium, provided the original work is properly cited.

[revised manuscript text omitted]

---

## Author Response (AR2)

November 21, 2020

Dear Rebecca Hodge,

On the behalf of co-authors, I would like to thank you for the thorough and constructive comments. They were very useful to improve the overall quality and readability of the manuscript. We have tried answering all the suggestion and comments made by you and revised the manuscript accordingly.

We have enclosed the following with this letter ;

— a detailed response

— a difference file showing the changes in the previous version of the manuscript in response to the comments

I hope this revised manuscript will be considered suitable for publication in Esurf.

Your truly

Kumar Gaurav

**Answer to Dr. Rebecca Hodge**

Text in black is the comments

Text in blue is the author's response

Line no: 30 It's possible to use these equations to explain two slightly different things. One is how channel properties at mean flow change between different locations along a river. The other is how, at a specific location, the properties of the channel change as discharge changes. Which are you referring to? Section 3 implies that is the former, in which case you need to specify which discharge you are referring to (mean/bankfull/flood).

Yes, in our case we are using the width-discharge regime equation to explain the channel properties at formative discharge between different locations along a river reach. The discharge that we are referring corresponds to bankfull discharge. Accordingly we have specified this in the revised manuscript.

Line no: 67 I'm not sure what you mean by 'unique' - specific to these particular rivers?

We have removed 'unique' from the sentence.

Line no: 69 By river width, do you mean the width of the water, or the width between the banks (water plus exposed sediment, typically identified as the edge of the floodplain vegetation)? This might be obvious to you, but I can imagine that different readers might assume

Our river width corresponds to the width of the water in a thread. In braided river, this is usually taken from margin to margin of an individual thread. As already explained in the manuscript, we consider braided threads as a collection of individual threads and we treat them separately to extract the wetted width observed from satellite images.

Line: 85 In Fig 1, I wouldn't say that the discharge looks constant in the unshaded region. Consider rewording.

Thank you for highlighting this. We have modified the sentence in the revised manuscript.

Line no: 89 Returning to my previous point, was Lacey comparing width at mean discharge between different locations, or variations in width with discharge at a specific location?

A quick answer to this question is regime channels. The measurements reported in Lacey 1930 and used by him to derive Lacey's law firstly come from studies performed on different Indian irrigation canals by Kennedy and Lindsey at the turn of the 20th century. The measurements reported are values measured for different "experiments" where controlled constant discharges where flowing through an irrigation channel and the corresponding hydraulic geometry measured. Lacey also compares these measurements to a small set of measurements performed on Egyptian canals. Then he shows that his law also works for regime data gathered from different rivers of India (the Ganga, Irrawady, Chenab etc). There is a very interesting discussion on the way engineers empirically tried to "fit" the Chenab river into a single stable channel that falls on the $W \propto Q^{0.5}$ line.

Line no: 104 Explain the implication in terms of later estimates of discharge - I think that using 0.3 means that your estimates are the lowest possible discharge for a given width?

It appears there are some misunderstandings here. Equation 4 predicts the width of a threshold channel to imposed discharge. According to this equation, width of a threshold channel scales as a square root of discharge. Parameters in the pre-factor depend on fluid and sediment properties. Some of these parameters such as water and sediment density, friction angle are approximately constant. However, in rivers, the other parameters such as the median grain size, turbulent friction coefficient and Shield parameter $(\theta_t)$ vary significantly. It has been found that the Shield parameter ranges between 0.03-0.3, depending on the Reynolds number on grain scale (Métivier et al., 2017). As we have already mentioned in the manuscript that Delorme et al. (2017), obtained an experimental value of Shield parameter 0.25 for silica grains of size 150 micron. As the typical grain-size of the Himalayan rivers vary between 100-300 micron, we have taken the upper value of $\theta_t = 0.3$ as a conservative estimate. Since this value is experimentally obtained for the grain size that is comparable to the sediment size of the Himalayan Foreland river, it provides more confidence. However for a similar discharge, using a lower value of $\theta_t$ (i.e; 0.1 or lesser) would lead to a slightly wider cross-section of a channel. Further, it is observed that, at a given discharge, width of a natural channel scales in a similar fashion to that of threshold channel. However, natural rivers are much wider (about a factor of 2 in our case ) than the threshold channel. Though using a lowest possible value of $\theta_t$ would not result to a closer match to the width of a natural channel. Therefore in this study, we have adjusted the pre-factor to best-fit the data points while keeping the theoretical exponent to establish a semi-empirical width-discharge curve. Later we use this curve to estimate discharge from the measurement of channel's width on remote sensing images.

Line no: 115 From this I assumed that you would be using the curve in Fig 2 in this paper, but I see later that this is not the case. This needs to be more clearly explained. (See also comment on page 12)

We are using the same curve shown in Fig 2 to estimate discharge from thread's width extracted from satellite images. This we have also explained in lines 117-121. We have clearly written that the threshold theory well predicts the exponent of the width-discharge relationship of the

Himalayan rivers but not the pre-factor. Threads of the Himalayan rivers are wider (about factor of 2) than the theoretical prediction. To go further we have adjusted the pre-factor to the data while keeping the theoretical exponent to establish a generalised semi-empirical regime curve for the Himalayan rivers. Finally, we use this "semi-regime curve" to estimate discharge from satellite images.

Line no: 116 Further details are needed. Explain which width (wetted or total channel width), and which discharge (mean/bankfull/flood).

Done. We have measured the wetted width of the threads using an Acoustic Doppler Current Profiler (ADCP) in the field. Most of our measurements were acquired during the period when rivers of the Himalayan Foreland usually flow at their bankfull discharge.

Line no: 127 include the pixel size of the different datasets

Done, we have included this in subsection 4.2 (width extraction). The spatial resolution of the IR band of Landsat image is 30 meters. For Sentinel-1 images we get a spatial resolution of 60 meters after the removal of speckle noise. We then resampled this to $(30 \times 30)$m square pixel.

Figure 2 caption: As with the text, you need to explain which width and discharge you are referring to.

Done

Line no: 184 Is this a manual or automated procedure? Is this one that you have developed, or adopted from elsewhere? You need to include more information about this in the main text.

This is an automated procedure that we have developed to measure the wetted width of individual threads from remote sensing images. This is already mentioned in section 4.2 (lines 144-146). We have also provided the detailed description of the algorithms that we have developed to split the water and non-water pixels from the satellite images (section 4.2) and eventually to measure channel width in Appendix B (Satellite image processing).

Line no: 210 Define what you mean by relative error.

Relative error is a measure of precision. We have calculated this by taking the ratio of the absolute error between manually and automatically measured width to automatically measured width of a thread. As suggested, we have added a line to define the relative error.

Line no: 219 It's not clear to me that the geometric mean is necessarily the most probable value. I would have said that the mode is the most probable value, as that is the value that the largest number of widths have.

We agree and restrict to write "most probable" width, instead we use geometric mean as a

"representative width". We have used geometric mean because it is less affected by extreme values in a skewed distribution.

Line no: 231 I thought from earlier that you were going to use the equation in Fig 2, but I see now that you can't use that because it contains grain size. This needs to be explained either earlier, or here. Somewhere you also need to explain how Gaurav et al established their regime relation

In the revised manuscript we have explained (line no: 117-123) how Gaurav et al has established the width-discharge regime curve for the Himalayan Foreland rivers. Yes, we have used equation 4 as explained in Fig 2. We have rearranged the equation 4 to estimate discharge from the measurement of width. To explain this let me rewrite how equation 6 was obtained from equation 4;

$$\frac{W}{d_s} = \left[ \frac{\pi}{\mu} \left( \frac{\theta_t(\rho_s - \rho_f)}{\rho_f} \right)^{0.25} \sqrt{\frac{3C_f}{2^{3/2}\mathcal{K}\left[1/2\right]}} \right] Q_*^{0.5} \tag{1}$$

where $Q_* = Q_w/(d_s^2\sqrt{gd_s})$ is the dimensionless water discharge. Please refer to the manuscript for the description of other parameters. The Himalayan River scales according to threshold theory (1), but not the pre-factor (in square bracket). Now we keep the exponent as predicted by theory and adjust the pre-factor to best-fit our data points. The resulting width-discharge curve reads;

$$\frac{W}{d_s} = \alpha \, Q_*^{0.5} \tag{2}$$

where $\alpha$ is the best-fit coefficient obtained from adjusting the pre-factor. This semi-empirical curve (Eqn. 2) has the theoretical exponent and empirical pre-factor.

Now we rearrange equation 2 to get Q;

$$W/d_s = \alpha \, (Q_w/d_s^2\sqrt{gd_s})^{0.5} \tag{3}$$

Finally we get;

$$Q_w = \left( \frac{W_m}{\alpha} \right)^2 \sqrt{(gd_s)}, \tag{4}$$

Line 232 It's not clear to me from this how you get a value for alpha if you only have the width measurements.

Already explained above. Also in the revised manuscript (line 240) we have defined '$\alpha$'. "This is the best-fit coefficient, an empirical value obtained from fitting the prefactor of the regime curve (Eq.4)"

Figure 9 Need to explain how error bars are calculated.

Explained in the caption.

Table 1 Need to explain how errors are calculated.

Explained in the caption.

Line no: 248 But in some cases there is a clear annual cycle, so the image does seem to better represent the monthly discharge. It would be useful to explore why the predicted discharge varies annually at some sites, and is constant at others. I assume that it is related to the shape of the channel, and hence how much the wetted width varies with discharge?

Thank you for highlighting this. We have written this in revised manuscript (line: 249-252). Though figure 9 shows annual cycle for Indus, Cheenab, and Teesta River, but it is not observed in the remaining rivers. As already explained in the discussion, these rivers are highly regulated (lines 296-300), and also discharge records for the Teesta at Kaunia (1969 - 1975) and at Anderson bridge (1965 - 1971) are available only for 6 and 7 years respectively (Table A2 in appendix). Similarly, the records for both Chenab and Indus rivers extend over 7 years only (1973 - 1979). These measurements are old and thus we refrain from commenting on the observed annual cycle. Further, we can also observe a strong declining trend of monthly discharges of the Indus, Chenab and Teesta rivers (figures C2 and C3 in appendix, in the manuscript). This validates that the hydrology of these rivers may have evolved since these records were established.

Line no: 270 This still doesn't quite explain to me why the wetted width doesn't vary with discharge in some channels. Doesn't it depend on the channel shape?

To answer this lets explain the cross-section geometry of a braided thread. Figure 1 illustrate the cross-section of a braided thread measured in the field using Acoustic Doppler Current Profiler (ADCP). This high resolution bathymetric profile enables us to identify the two different threads (see the velocity cells) separated by a region of very shallow flow depth (submerged bar). As a consequence, discharge is not carried through the width that is observed from the top, but only through the region where we observe the velocity cells. It is important to notice here that the geometry of this thread must have been set at the formative discharge. During low flow usually threads maintain their flows without modifying the existing geometry.

Currently satellite images allow us to measure the top width of water surface, this is perhaps one of the reasons why our estimated of average monthly discharge is mostly overestimated then the in-situ value and does not show much variability.

Line no: 278 I'm confused by this section. You seem to be arguing that the measured width, and

[Figure]

Figure 1: Velocity profile measured using and ADCP across a braided thread of the Kosi River in the Himalayan Foreland. Red horizontal line with arrow is to illustrate the top water surface. Colors with different intensities show the magnitude of velocity (m/s)

hence discharge, doesn't vary between months. I don't agree with this, because the predicted discharge clearly does vary between months at some sites in Fig 9. Furthermore, Fig C2 and C3 seem to show that your data do reproduce observed differences between months. I'm not sure what Fig C1 is meant to show.

Justification to both the points are explained above. To comment on the observed variability in some stations (Indus, Cheenab, and Teesta) we modified the discussion section in the revised manuscript.

Line no: 283 Again, I don't agree with this interpretation of Fig 9

Now we have modified this sentence and write, except for the rivers (Indus, Chenab and Teesta) we have old and limited records of in-situ discharge. "We observe that the estimated discharges from images are nearly constant during the monsoon period, with only small fluctuations around their mean" We think this justifies our interpretation of fig 9.

line no: 291 Are the images and the discharge data from the same time period? The datasets referred to in the methods covered a wide range of time periods. Please clarify here.

The in-situ discharge data and images are not from the same time period. This is already mentioned in the manuscript in sub-section 4.1 and Table A2 in the Appendix.

Line no: 317 Provide a bit more information in the main text about how the different estimates compare

Done

Line no: 323 Any river, or just those above a certain width?

Since our width-discharge regime curve is established from numerous channels of different sizes, we believe it can be used to approximate the formative discharge of any rivers of the Himalayan Foreland.

Line no: 325 Which measurements? The new data in this paper are from all months. You need to make it clear that this paragraph is all referring to previous work.

We have explained the measurements and accordingly modified the sentences to make it more clear.

Line no: 328 formative or instantaneous?

Formative

Line no: 339 You took width measurements from a range of different data sources (images/ASAR), but have not compared them. Are some data sources more reliable than others?

This is one of the limitations, we could not compare our threads width extracted from Landsat and SAR images. Since the temporal resolution of both the images are different we could not find common acquisition dates for both the datasets. Further it doesn't make much sense to compare the measurements obtained from images of two different dates. As already discussed in the manuscript (129-135), that SAR sensors can acquire uninterrupted image of the Earth's surface even during the bad weather conditions. This is an important dataset that allows us to monitor rivers during the monsoon period, where optical images are affected by cloud cover and strong rainfall.

Appendix For each location in C2 and C3, use the same vertical scale

Done

**Coupling threshold theory and satellite image derived channel width to estimate the formative discharge of Himalayan Foreland rivers.**

Kumar Gaurav[1], François Métivier[2], AV Sreejith[3], Rajiv Sinha[4], Amit Kumar[1], and Sampat Kumar Tandon[1]

[1]Indian Institute of Science Education and Research, Bhopal,462066, M.P, India
[2]Institute de Physique du Globe de Paris, 1 Rue Jussieu, 75005 Paris cedex 05, France
[3]School of Mathematics and Computer Science, Indian Institute of Technology, Goa, 403401, Goa, India
[4]Department of Earth Sciences, Indian Institute of Technology, Kanpur, 208016 UP, India

**Correspondence:** K.Gaurav (kgaurav@iiserb.ac.in)

**Abstract.** We propose an innovative methodology to estimate the formative discharge of alluvial rivers from remote sensing images. This procedure involves automatic extraction of the width of a channel from Landsat Thematic Mapper, Landsat 8, and Sentinel-1 satellite images. We translate the channel width extracted from satellite images to discharge by using a width-discharge regime curve established previously by us for the Himalayan Rivers. This regime curve is based on the threshold theory, a simple physical force balance that explains the first-order geometry of alluvial channels. Using this procedure, we estimate the discharge of six major rivers of the Himalayan Foreland: the Brahmaputra, Chenab, Ganga, Indus, Kosi, and Teesta rivers. Except highly regulated rivers (Indus and Chenab), our estimates of the discharge from satellite images can be compared with the mean annual discharge obtained from historical records of gauging stations. We have shown that this procedure applies both to braided and single-thread rivers over a large territory. Further our methodology to estimate discharge from remote sensing images does not rely on continuous ground calibration.

**Keywords:** Himalayan Foreland; regime curve; threshold theory; formative discharge

*Copyright statement.* This is an open access article under the terms of the Creative Commons Attribution License, which permits use, distribution and reproduction in any medium, provided the original work is properly cited.

[revised manuscript text omitted]

---

## Author Response (AR3)

December 08, 2020

Dear Rebecca Hodge,

On the behalf of co-authors, I would like to thank you for the constructive comments. They were very useful, we have tried answering all the suggestion and comments made by you in the revised the manuscript.

We have enclosed the following with this letter;

- a response file
- a difference file showing the changes in the previous version of the manuscript in response to the comments

I hope this revised manuscript will be considered suitable for publication in Esurf.

Your truly

Kumar Gaurav

**Answer to Dr. Rebecca Hodge**

Text in black is the comments Text in blue is the author's response

Thanks for your detailed response to the review. I realise that a number of my comments were because of confusion about whether or not you were using grain size data in your calculations, as the data were not previously mentioned. I think that this is an important point to make in the methods, as readers might assume that a remote sensing technique does not require field data. Your explanation about what sets the observed width is an important addition, but which was better described in the response to reviews that in the paper.

Thank you for highlighting this. In the revised manuscript we have clarified both the points. In the material and method section (lines: 148-151) we have mentioned the grainsize data. Also in the discussion section, we have explained the cross-sectional width of a braided river seen from the top. In support of this we have included a cross-section profile measured in the field using an ADCP in Appendix-1 (Fig.C1) to explain why a channel width observed from satellite images is wider than the actual width.

**Coupling threshold theory and satellite image derived channel width to estimate the formative discharge of Himalayan Foreland rivers.**

Kumar Gaurav1, François Métivier2, AV Sreejith3, Rajiv Sinha4, Amit Kumar1, and Sampat Kumar Tandon1

1Indian Institute of Science Education and Research, Bhopal,462066, M.P, India
2Institute de Physique du Globe de Paris, 1 Rue Jussieu, 75005 Paris cedex 05, France
3School of Mathematics and Computer Science, Indian Institute of Technology, Goa, 403401, Goa, India
4Department of Earth Sciences, Indian Institute of Technology, Kanpur, 208016 UP, India

Correspondence: K.Gaurav (kgaurav@iiserb.ac.in)

**Abstract.** We propose an innovative methodology to estimate the formative discharge of alluvial rivers from remote sensing images. This procedure involves automatic extraction of the width of a channel from Landsat Thematic Mapper, Landsat 8, and Sentinel-1 satellite images. We translate the channel width extracted from satellite images to discharge by using a width-discharge regime curve established previously by us for the Himalayan Rivers. This regime curve is based on the threshold

- 5 theory, a simple physical force balance that explains the first-order geometry of alluvial channels. Using this procedure, we estimate the discharge of six major rivers of the Himalayan Foreland: the Brahmaputra, Chenab, Ganga, Indus, Kosi, and Teesta rivers. Except highly regulated rivers (Indus and Chenab), our estimates of the discharge from satellite images can be compared with the mean annual discharge obtained from historical records of gauging stations. We have shown that this procedure applies both to braided and single-thread rivers over a large territory. Further our methodology to estimate discharge
- 10 from remote sensing images does not rely on continuous ground calibration.Keywords: Himalayan Foreland; regime curve; threshold theory; formative discharge

*Copyright statement.* This is an open access article under the terms of the Creative Commons Attribution License, which permits use, distribution and reproduction in any medium, provided the original work is properly cited.

[revised manuscript text omitted]